# The balancing principle for parameter choice in distance-regularized domain adaptation

Werner Zellinger[1,*]    Natalia Shepeleva[1]    Marius-Constantin Dinu[2,3]

Hamid Eghbal-zadeh[4,5]    Duc Hoan Nguyen[6]    Bernhard Nessler[2]

Sergei V. Pereverzyev[6]    Bernhard A. Moser[1]

[1]Software Competence Center Hagenberg GmbH
[2]Institute for Machine Learning, Johannes Kepler University Linz
[3]Dynatrace Research
[4]Institute of Computational Perception, Johannes Kepler University Linz
[5]LIT AI Lab, Johannes Kepler University Linz
[6]Johann Radon Institute for Computational and Applied Mathematics, Austrian
Academy of Sciences
[*]`werner.zellinger@scch.at`

## Abstract

We address the unsolved algorithm design problem of choosing a justified regularization parameter in unsupervised domain adaptation. This problem is intriguing as no labels are available in the target domain. Our approach starts with the observation that the widely-used method of minimizing the source error, penalized by a distance measure between source and target feature representations, shares characteristics with regularized ill-posed inverse problems. Regularization parameters in inverse problems are optimally chosen by the fundamental *principle of balancing* approximation and sampling errors. We use this principle to balance learning errors and domain distance in a target error bound. As a result, we obtain a theoretically justified rule for the choice of the regularization parameter. In contrast to the state of the art, our approach allows source and target distributions with disjoint supports. An empirical comparative study on benchmark datasets underpins the performance of our approach.

## 1   Introduction

*Domain adaptation* uses the knowledge in a *source* domain to improve the performance of an algorithm on a related *target* domain [1]. In particular, domain adaptation tackles domain shifts in machine learning applications: Medical diagnostic systems should be adapted to new physical human variations; Industrial quality inspection systems should be accurate for new products; Self-driving cars should be able to adapt to new geographical environments and weather conditions. In this work, we focus on *unsupervised domain adaptation* where labels are only available in the source domain.

There are mainly two types of approaches for unsupervised domain adaptation: *importance weighting* [2, 3, 4, 5, 6, 7] and *feature representation learning* [8, 9, 10, 11, 12, 13]. In this work, we focus on feature representation learning which goes beyond classical importance weighting by allowing a target distribution with support outside of the source distribution. The core idea behind feature representation learning approaches is to map the data into a new feature space where the source and target

35th Conference on Neural Information Processing Systems (NeurIPS 2021).

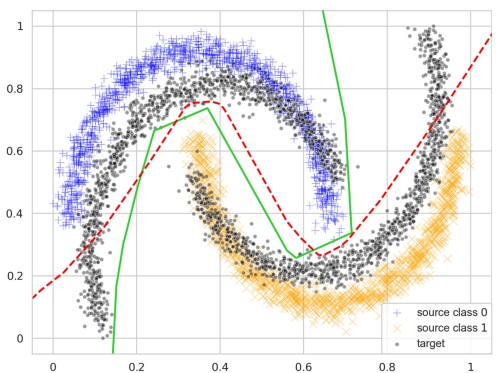 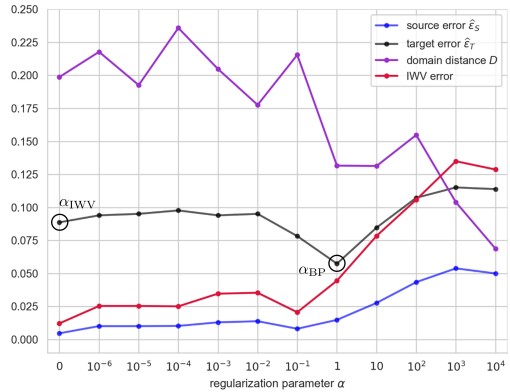

Figure 1: Unsupervised domain adaptation on Transformed Moons. Left: Target data (black dots) partially outside of the support of the source data (blue +, orange ×). The common assumption of bounded density ratio is violated in large regions. In contrast, all our assumptions are satisfied. Our method identifies the best parameter of a domain adaptation algorithm [16] (green solid) which improves training on source data only (red dashed). Right: Regularization parameter (x-axis) which penalizes a distance [25] (purple) leading to models with different source error (blue) and target error (black). Importance weighted validation (IWV) shows the smallest error (red) for models without domain adaptation ($\alpha_{\mathrm{IWV}} = 0$). In contrast, our approach identifies the optimal parameter ($\alpha_{\mathrm{BP}} = 1$).

data representations appear similar, and where enough information is preserved for prediction [14]. The similarity is often realized by *regularization using distance measures* between source and target representations [15, 16, 17, 18, 19, 20, 21]. However, the performance of such methods crucially depends on the choice of the regularization parameter which penalizes the distance. The problem we investigate in this work is to choose this parameter, which is sophisticated without any target labels.

While remarkable theoretical results have been achieved which quantify the generalization ability of domain adaptation models [8, 22, 19, 20, 21], the choice of the regularization parameter which is crucial for finding such models has not systematically been addressed. Even though some parameter choice strategies exist, they are either purely heuristically driven or very limited by their assumptions [23]. Typical approaches are fixing the regularization parameters [12], minimizing the source error [16], balancing the source error and a distance [17], multiplying a fixed weighting parameter (e.g. 1 in [16]) by a heuristic schedule value that increases during training, or, (importance) weighting the input samples by the ratio between target and source density [3, 24, 23]. One common problem shared among all these approaches is that they all can fail if the density ratio is unbounded. Such unbounded density ratio is typical for many of the high dimensional problems considered in machine learning [19], e.g. see Figure 1. Besides the aforementioned issues, the lack of principled strategies for parameter choice causes misinterpretations in the ranking of domain adaptation methods which are traditionally compared by performance, while often relying on different parameter choice strategies.

In this work, we propose a principled method for choosing distance-penalizing parameters of feature representation learning approaches for unsupervised domain adaptation. Our approach starts with the observation that the distance-regularization setting of domain adaptation shares characteristics with regularized ill-posed inverse problems (see Table 1). In inverse problems, the regularization parameter can be optimally chosen by the fundamental *balancing principle* which optimizes an approximation-sample (bias-variance) trade-off [26, 27, 28]. We apply this principle for balancing *domain distance* and *learning errors* of target error bounds. In particular, we approach the problem of non-computable terms in the target error bound by a new algorithmic criterion for approximating the value of balance. We call our method the Balancing Principle for Domain Adaptation (BPDA).

The BPDA is general in the sense that it can be applied based on different target error bounds, e.g. on [8, 22, 19, 20, 21]. To the best of our knowledge, the BPDA is the first principled method for parameter choice in unsupervised domain adaptation that allows an unbounded ratio between target and source density. We provide a bound on the generalization error of the best model corresponding to the parameter chosen by the BPDA. Finally, we empirically investigate the behavior of the BPDA based on two target error bounds, different domain adaptation methods and benchmark datasets. Our

results show that the BPDA outperforms or is on par with the state of the art on the problem of choosing the regularization parameter, on several domain adaptation methods; applied on different datasets.

## 2 Summary of results

**Notation**  Let $\mathcal{X} \subset \mathbb{R}^n$ be an *input space* and $\mathcal{Y}$ be a discrete *label space*. Following the classical setting of unsupervised domain adaptation [25], we consider two datasets: A *source dataset* $(\mathbf{x}, \mathbf{y}) = ((x_1, l_S(x_1)), \ldots, (x_s, l_S(x_s))) \in (\mathcal{X} \times \mathcal{Y})^s$ with *inputs* $x_1, \ldots, x_s$ independently drawn according to some source distribution (Borel probability measure) $p_S$ on $\mathcal{X}$ and labeled according to some *labeling function*[1] $l_S : \mathcal{X} \to \mathcal{Y}$, and, an unlabeled *target* dataset $\mathbf{x}' = (x'_1, \ldots, x'_t) \in \mathcal{X}^t$ with elements independently drawn according to some target distribution $p_T$ on $\mathcal{X}$. Throughout this work, we focus on loss functions $L : \mathcal{Y} \times \mathcal{Y} \to [0, \infty)$ which satisfy $L(y, y) = 0$. For example consider the 0-1 loss $L(y_1, y_2) := \mathbb{1}[y_1 \neq y_2]$, where $\mathbb{1}[P]$ is 1 iff the predicate $P$ is true and 0 otherwise, and the quadratic loss function $L(y_1, y_2) := |y_1 - y_2|^2$. We denote the source error by $\varepsilon_S(f) = \varepsilon_S(f, l_S)$ with *cross-error* defined as $\varepsilon_S(f, g) := \mathbb{E}_{x \sim p_S}[L(f(x), g(x))]$ and its empirical sample estimate by $\widehat{\varepsilon}_S(f) = \widehat{\varepsilon}_S(f, l_S)$ with $\widehat{\varepsilon}_S(f, g) := \sum_{i=1}^s L(f(x_i), g(x_i))$. We denote the analogously defined target error by $\varepsilon_T(f)$, target cross-error by $\varepsilon_T(f, g)$ and its empirical sample estimate by $\widehat{\varepsilon}_T(f)$ with empirical cross-error $\widehat{\varepsilon}_T(f, g)$. Throughout this work, we focus on target cross-errors $\varepsilon_T(f, g)$ which satisfy the triangle inequality.

**Learning setup**  In this work, we focus on feature representation learning algorithms for domain adaptation. These approaches aim at finding two learning models: A *representation* mapping $\phi \in \Phi \subset \{\phi : \mathcal{X} \to \mathcal{R}\}$ into some representation space $\mathcal{R} \subset \mathbb{R}^m$ and a *classifier* $g \in \mathcal{G} \subset \{g : \mathcal{R} \to \mathcal{Y}\}$. Loosely speaking, the aim is to find a mapping $\phi$ under which the source representations $\phi(\mathbf{x}) := (\phi(x_1), \ldots, \phi(x_s))$ and the target representations $\phi(\mathbf{x}') := (\phi(x'_1), \ldots, \phi(x'_t))$ appear similar, and, at the same time, enough information is preserved for prediction [14] by $g(x)$. A common approach to realize this aim is to solve the following objective function [19]

$$\min_{g \in \mathcal{G}, \phi \in \Phi} \widehat{\varepsilon}_S(g \circ \phi) + \alpha \cdot d(\phi(\mathbf{x}), \phi(\mathbf{x}')) \tag{1}$$

where $d$ is a distance measure between source and target representations and $\alpha \in [0, \infty)$ is a parameter[2]. Good choices for $d$ in Eq. (1) have been identified to be the Wasserstein distance [29, 30], the Maximum Mean Discrepancy [31, 32], moment distances [17, 33, 18, 34, 35, 36], adversarially learned distances [16, 37] and other measures of divergence [38, 39, 19, 20].

**Problem**  For some $\alpha \in [0, \infty)$, let $g_\alpha \circ \phi_\alpha$ denote the minimizer of Eq. (1). Given an increasing sequence of parameters $\alpha_1, \ldots, \alpha_w \in [0, \infty)$ with $\alpha_1 = 0$, the problem studied in this work is to choose the parameter $\alpha$ in the sequence $\alpha_1, \ldots, \alpha_w$ with the lowest target error $\varepsilon_T(g_\alpha \circ \phi_\alpha)$.

**Approach**  Our approach consists in minimizing a target error bound which satisfies the form

$$\varepsilon_T(g_\alpha \circ \phi_\alpha) \leq D(\alpha) + E(\alpha) \tag{2}$$

where $D(\alpha)$ gives a notion of *domain distance* (cf. [25]) by quantifying a distance between source and target data representations and $E(\alpha)$ comprises different *learning errors*. We assume that $E(\alpha)$ is bounded by some constant $B > 0$. The general form in Eq. (2) is satisfied by many error bounds [8, 22, 20, 21, 34] which all can be taken as a basis for our approach (more detailed examples are provided in Section 3 and Section 4). One problem that complicates the minimization of these error bounds is that all of them contain terms that are not computable due to the lack of target data. That is, $E(\alpha)$ cannot be directly estimated. The BPDA overcomes this problem by a new criterion for estimating the value of balance between the normalized terms $E(\alpha)$ and $D(\alpha)$. The BPDA is detailed in Algorithm 1.

**Properties of Algorithm 1**  The BPDA has the following striking properties.

- The BPDA is a general procedure which can be instantiated by any error bound of the form in Eq. (2). See Section 4 and Section 5 for its application based on two different target error bounds [25, 20].

---

[1]For simplicity, we use labeling functions instead of the more general concept of conditional distributions.
[2]For simplicity we omit further regularization of $\phi$ and $g$.

- In contrast to state-of-the-art methods, the BPDA does not assume a target labeling function $l_T$ that is equal to the source labeling function $l_S$ (covariate-shift assumption) and it does not assume that the ratio between target and source density is bounded (sample selection bias assumption). See the supplementary material for a discussion of covariate-shift violations.

- The learning model $g_{\alpha_{\mathrm{BP}}} \circ \phi_{\alpha_{\mathrm{BP}}}$ identified by the BPDA satisfies a generalization bound, see Section 4. If the learning errors term $E(\alpha)$ is non-decreasing, then the target error of $g_{\alpha_{\mathrm{BP}}} \circ \phi_{\alpha_{\mathrm{BP}}}$ is only a constant factor away from the minimum $\min_{\alpha \in [0,\infty)} D(\alpha) + E(\alpha)$ of the instantiation bound in Eq. (2).

---

**Algorithm 1:** Balancing principle for domain adaptation (BPDA)

---

**Input** : Increasing sequence of parameters $\alpha_1, \dots, \alpha_w \in [0,\infty)$ with $\alpha_1 = 0$ and minimizers $f_1 := g_{\alpha_1} \circ \phi_{\alpha_1}, \dots, f_w := g_{\alpha_w} \circ \phi_{\alpha_w}$ of Eq. (1).

**Output** : Parameter $\alpha_{\mathrm{BP}}$ solving the problem above.

**Initialization** : $S = \{\}$

**for** $i = 1, \dots, w$ **do**

    Compute empirical cross-error $\widehat{\varepsilon}_T(f_i, f_j)$ and domain distance $D(\alpha_j)$ for all $j = 1, \dots, i - 1$.

    **if** $\widehat{\varepsilon}_T(f_i, f_j) \leq D(\alpha_j) \left(2 + \frac{2B}{D(0)}\right)$ *for all* $j = 1, \dots, i - 1$ **then**

        |   $S := S \cup \{\alpha_i\}$

    **end**

**end**

**return** : $\alpha_{\mathrm{BP}} := \max S$

---

Besides the properties above, the BPDA outperforms or is on par with the state of the art, on the problem of choosing the regularization parameter, on several domain adaptation methods; applied on different datasets, see Section 5.

**Related work** Approaches which follow Eq. (1) are sometimes interpreted as learning *domain-invariant* representation. Note that the *minimization* of $d(\phi(\mathbf{x}), \phi(\mathbf{x}'))$ in Eq. (1) to achieve $\phi(\mathbf{x}) = \phi(\mathbf{x}')$ differs from the conception as *regularization* [40, 41]. In fact, minimization of the distance means unjustified over-penalization which might lead to deteriorated performance [19]. Interestingly, our interpretation as regularization problem opens up a powerful toolbox of mathematical techniques. Our approach takes up the technique of balancing stability and approximation in regularized ill-posed inverse problems. The balancing principle has its origins in [26] devoted to nonparametric regression estimation and has been introduced in the context of ill-posed problems [27] and in supervised learning with kernels [42]. Following this line of research, we propose to apply the mathematical techniques underlying the balancing principle in the context of domain adaptation. The most related principled parameter choice methods in the context of unsupervised domain adaptation are importance-weighted cross-validation [3] and its extensions [24, 23]. In contrast to these methods, our method is not restricted by the assumption of a bounded ratio between target and source density. One empirically driven method which is related to ours is [17, 43] which aims at balancing the source error and a distance between source and target data representations. However, this method is not theoretically justified as it ignores the minimal combined error of a classifier on representations as defined in [8]. Nevertheless, if the minimal combined error is negligibly small, our results provide a theoretical explanation of the principles underlying the success of [17, 43]. Our method relies on bounds on the target error such as e.g. [8, 22, 19, 20, 21, 34].

## 3  Preliminaries

**Sampling error bound** Throughout this work, we generically denote an upper bound on the sampling error by $\eta_{t,\mathcal{G},\delta} \in [0,\infty)$, which is assumed to hold true with probability at least $1 - \delta$:

$$|\varepsilon_T(f,g) - \widehat{\varepsilon}_T(f,g)| \leq \eta_{t,\mathcal{G},\delta} \qquad (3)$$

The bound $\eta_{t,\mathcal{G},\delta}$ depends on the sample size $t$, the function class $\mathcal{G}$ and the constant $\delta$, and, it is assumed to satisfy $\eta_{t,\mathcal{G},\delta} \to 0$ for $t \to \infty$.

**Two accompanying target error bounds** In the seminal works [8, 25] binary classification $\mathcal{Y} := \{0, 1\}$ with 0-1 loss is considered and it is shown that the following bound holds for all symmetric function classes $\mathcal{G}$ ($g \in \mathcal{G} \implies 1 - g \in \mathcal{G}$), $\phi \in \Phi$, $g \in \mathcal{G}$ and datasets of equal size $s = t$ with probability at least $1 - \delta$:

$$\varepsilon_T(g \circ \phi) \leq \varepsilon_S(g \circ \phi) + \lambda_{\mathcal{G}}(\phi) + \widehat{d}_{\mathcal{G} \triangle \mathcal{G}}(\phi(\mathbf{x}), \phi(\mathbf{x}')) + \eta_{t, \mathcal{G}, \delta} \tag{4}$$

where

$$\widehat{d}_{\mathcal{G} \triangle \mathcal{G}}(\mathbf{x}, \mathbf{x}') := 2 \left( 1 - \min_{f, f' \in \mathcal{G}} \left[ \frac{1}{s} \sum_{i=1}^{s} \mathbb{1}[f(x_i) = f'(x_i)] + \frac{1}{s} \sum_{i=1}^{s} \mathbb{1}[f(x_i') \neq f'(x_i')] \right] \right) \tag{5}$$

is the empirical $\mathcal{G} \triangle \mathcal{G}$-*divergence* [44, 25] and $\lambda_{\mathcal{G}}(\phi) := \inf_{f \in \mathcal{G}}(\varepsilon_S(f \circ \phi) + \varepsilon_T(f \circ \phi))$ is the minimum possible *combined error* determined by the application of $\mathcal{G}$. Using Eq. (3), the source error can be further upper bounded by the empirical source error. However, the term $\lambda_{\mathcal{G}}(\phi)$ cannot be estimated based on given datasets as it depends on the unknown labeling function $l_T$.

In [20], the bound in Eq. (4) is generalized to multiple classes $\mathcal{Y} := \{1, \ldots, k\}$ and scoring functions $\mathcal{H} \subset \{h : \mathcal{R} \to \mathbb{R}^k\}$, where the output on each dimension indicates the confidence of prediction. For some $h \in \mathcal{H}$, let us denote by $g_h : \mathcal{R} \to \mathcal{Y}, x \mapsto \arg\max_{i \in \{1, \ldots, k\}} h^{(i)}(\phi(x))$ with $h^{(i)}(z)$ being the $i$-th component of $h(z)$. Let further $\varepsilon_T$ denote the target error based on 0-1 loss. Then, with probability at least $1 - \delta$

$$\varepsilon_T(g_h \circ \phi) \leq \varepsilon_S^{(\rho)}(h \circ \phi) + \lambda_{\mathcal{H}}(\phi) + \widehat{d}_{h, \mathcal{H}}^{(\rho)}(\phi(\mathbf{x}), \phi(\mathbf{x}')) + \eta_{s+t, \mathcal{H}, \delta} \tag{6}$$

where $\varepsilon_S^{(\rho)}(h) := \int_{\mathcal{X}} \Lambda_\rho \circ \rho_h(x, l_S) \, dp_S(x)$ with $\rho_h(x, l_S) := \frac{1}{2}(h^{(l_S(x))}(x) - \max_{y \neq l_S(x)} h^{(y)}(x))$ and $\Lambda_\rho(x)$ being $1 - x/\rho$ if $0 \leq x < \rho$, 0 if $\rho \leq x$ and 1 otherwise. Further, the target terms $T, p_T, l_T$ are defined analogously to $S, p_S, l_S$, respectively; the combined error is defined by $\lambda_{\mathcal{H}}(\phi) := \inf_{h \in \mathcal{H}}(\varepsilon_S^{(\rho)}(h \circ \phi) + \varepsilon_T^{(\rho)}(h))$, and the *empirical margin disparity discrepancy* (MDD) is given by

$$\widehat{d}_{h, \mathcal{H}}^{(\rho)}(\mathbf{x}', \mathbf{x}) := \max_{h' \in \mathcal{H}} \left[ \frac{1}{s} \sum_{i=1}^{s} \Lambda_\rho \circ \rho_{h'}(x_i, g_h) - \frac{1}{t} \sum_{i=1}^{t} \Lambda_\rho \circ \rho_{h'}(x_i', g_h) \right] \tag{7}$$

Similarly to Eq. (4), the source error can be further estimated by an empirical error and the combined error $\lambda_{\mathcal{H}}(\phi)$ cannot be estimated based on given data.

**Balancing principle for regularized inverse problems** Let $\mathcal{H}$ and $\mathcal{K}$ be two Hilbert spaces and $V : \mathcal{H} \to \mathcal{K}$ be a linear operator. Then, the linear *inverse problem* associated to a given *datum* $g \in \mathcal{K}$ is to find some function $f$ satisfying $Vf = g$, see e.g. [45, 46, 40, 41] and references therein. In general, the above problem is ill-posed, i.e. a solution does either not exist, is not unique or does not depend continuously on $g$. Existence and uniqueness can be approached by using the the following minimizer as approximation of $f$

$$f_{\mathcal{H}} \in \arg\min_{f \in \mathcal{H}} \|Vf - g\|_{\mathcal{K}}^2$$

However, especially in the case of a noisy operator $\widehat{V}$ and noisy data $\widehat{g}$, continuous dependency on data becomes an important issue which can be restored using Tikhonov regularization [45]

$$f_{\alpha} \in \arg\min_{f \in \mathcal{H}} \|\widehat{V}f - \widehat{g}\|_{\mathcal{K}}^2 + \alpha \|f\|_{\mathcal{H}}^2 \tag{8}$$

In many cases, probabilistic bounds on the error can be proven

$$\|f_\alpha - f_{\mathcal{H}}\|_{\mathcal{H}} \leq S(\alpha) + A(\alpha) \tag{9}$$

where $S(\alpha)$ is called *sampling error* made by considering noisy approximations $\widehat{V}$ and $\widehat{g}$ of $V$ and $g$, respectively, and, $A(\alpha)$ is called *approximation error* originating from adding the regularizer $\|f\|_{\mathcal{H}}^2$. Commonly, $S(\alpha)$ decreases while $A(\alpha)$ increases with increasing $\alpha$. We refer to [42] for detailed examples. The fact that $S(\alpha)$ decreases and $A(\alpha)$ increases motivates the so-called *balancing principle* which aims at computing $\alpha^*$ such that $S(\alpha^*) = A(\alpha^*)$ [26, 27, 42, 41, 28]. As a result, the balancing principle provides a procedure for approximating $\alpha^*$ without having access to the values of $A(\alpha)$. The balancing principle obtains optimal error rates in many settings [47, 48, 49, 42, 41, 28].

# 4 Balancing principle for domain adaptation

In the following, we present the mathematical foundations of Algorithm 1 in two steps. In a first step, in Subsection 4.1, we state our assumptions and detail our idea of choosing the value of balance between terms in a target error bound. In a second step, in Subsection 4.2, we propose an algorithmic criterion, the *balancing principle estimate* for approximating the theoretical choice of the balancing value without target labels. Based on this criterion, we explain why we expect Algorithm 1 to be accurate.

## 4.1 Balancing terms in a target error bound

**Assumptions**   In the following, let $g_\alpha \circ \phi_\alpha$ be a minimizer of Eq. (1) and assume that a target error bound holds which is of the form given by Eq. (2). Our approach is based on the plausible assumptions (a) that the function $\alpha \mapsto E(\alpha)$ is continuous and bounded by some constant $B > 0$, and (b) that the function $\alpha \mapsto D(\alpha)$ is continuous, non-increasing and non-degenerate, i.e. $D(0) > 0$. We further make the technical assumption (c) that $\lim_{\alpha \to \infty} D(\alpha)/D(0) < \sup_{\alpha \in [0,\infty)} E(\alpha)/B$. Note that (c) is satisfied in most standard cases, where $D(\alpha) \to 0$ for $\alpha, t \to \infty$. For example consider the two accompanying target error bounds in Section 3 as discussed at the end of this section.

**Bridging regularized inverse problems and domain adaptation**   Under the assumptions above, our domain adaptation learning setup in Section 2 shares characteristics with the setting of regularized inverse problems as described in Section 3. Indeed, the Tikhonov regularizer $\|f\|_{\mathcal{H}}^2$ in Eq. (8) is similarly applied as the distance-regularizer $d(\phi(\mathbf{x}), \phi(\mathbf{x}'))$ in Eq. (1). In addition, error bounds for inverse problems as given by Eq. (9) show a similar form as target error bounds following Eq. (2). The sampling error $S(\alpha)$ in Eq. (9) decreases similarly to the domain distance in Eq. (2) and the approximation error $A(\alpha)$ in Eq. (9) cannot be estimated similarly to the learning errors $E(\alpha)$ in Eq. (2). However, in the domain adaptation setting, the term $E(\alpha)$ does not necessarily increase. We approach this issue by considering the *least non-decreasing majorant* of $E(\alpha)$.

**Definition 1** (Least non-decreasing majorant [50]). *The least non-decreasing majorant of $E(\alpha)$ is given by $\overline{E}(\alpha) := \sup_{\beta \in [0,\alpha]} E(\beta)$.*

Further upper bounding Eq. (2) by

$$\varepsilon_T(g_\alpha \circ \phi_\alpha) \leq D(\alpha) + E(\alpha) \leq D(\alpha) + \overline{E}(\alpha) \tag{10}$$

results in a form which satisfies all properties needed to apply the balancing principle, see Table 1.

Table 1: Correspondences between regularized inverse problems and domain adaptation which allow to apply the balancing principle. See Section 3 for details on the inverse problem setting.

| Tikhonov-regularized inverse problem | Distance-regularized domain adaptation |
|:---:|:---:|
| $f_\alpha \in \arg\min_{f \in \mathcal{H}} \|\widehat{V}f - \widehat{g}\|_{\mathcal{K}}^2 + \alpha \|f\|_{\mathcal{H}}^2$ | $g_\alpha \circ \phi_\alpha \in \arg\min_{g \in \mathcal{G}, \phi \in \Phi} \widehat{\varepsilon}_S(g \circ \phi) + \alpha\, d(\phi(\mathbf{x}), \phi(\mathbf{x}'))$ |
| $\|f_\alpha - f_{\mathcal{H}}\|_{\mathcal{H}} \leq S(\alpha) + A(\alpha)$ | $\varepsilon_T(g_\alpha \circ \phi_\alpha) \leq D(\alpha) + E(\alpha)$ |
| decreasing sampling error $S(\alpha)$ | decreasing domain distance $D(\alpha)$ |
| increasing approximation error $A(\alpha)$ | bounded learning errors $E(\alpha)$ |
| $A(\alpha)$ not estimable | $E(\alpha)$ not estimable |
| balance $A(\alpha^*) = S(\alpha^*)$ | balance $\frac{D(\alpha^*)}{D(0)} = \frac{\overline{E}(\alpha^*)}{B}$ |

**Balancing value for domain adaptation**   Having identified the shared characteristics between regularized inverse problems and domain adaptation, we now apply the techniques underlying the balancing principle to domain adaptation. We define the *balancing value* $\alpha^*$ as achieving

$$\frac{D(\alpha^*)}{D(0)} = \frac{\overline{E}(\alpha^*)}{B} \tag{11}$$

The normalizing factor $1/D(0)$ on the left-hand side of Eq. (11) and the factor $1/B$ ensure that the two curves $D(\alpha)/D(0)$ and $\overline{E}(\alpha)/B$ intersect. In particular, the existence of $\alpha^*$ follows from the

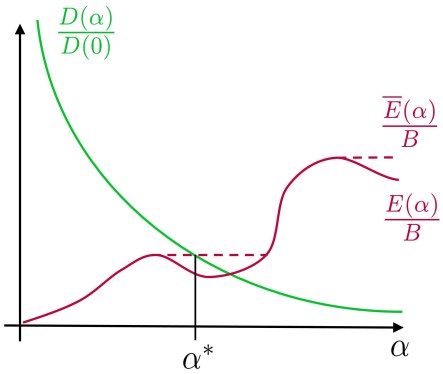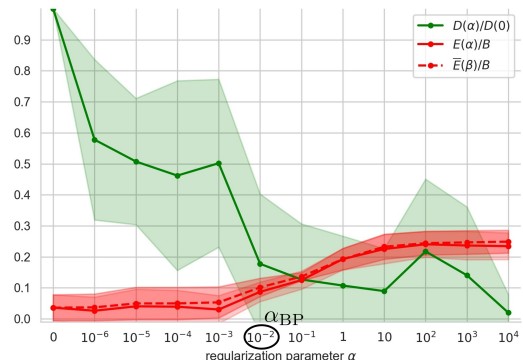

Figure 2: Left: The BPDA in Algorithm 1 overcomes the problem of the unknown learning errors term $E(\alpha)$ by approximating $\alpha^*$ which balances $D(\alpha)/D(0)$ (green) and the least non-decreasing majorant $\overline{E}(\alpha)/B$ (red dashed) of $E(\alpha)/B$ (red). Right: Average and standard deviation over 10 repetitions of estimated learning errors $E$ and the domain distance $D$ of the accompanying target error bound Eq. (4) for the models computed by Eq. (1) with the Maximum Mean Discrepancy [51] as distance. The BPDA chooses the value $\alpha_{\mathrm{BP}} = 10^{-2}$ near the estimated balancing value.

assumptions (a)–(c) above. See Figure 2 for an illustration. Algorithm 1 approximates $\alpha^*$. If $E(\alpha)$ is non-decreasing and the bound in Eq. (10) holds with equality, then the rate of the target error $\varepsilon_T(g_{\alpha^*} \circ \phi_{\alpha^*})$ is optimal, i.e. $\varepsilon_T(g_{\alpha^*} \circ \phi_{\alpha^*})$ is only a constant factor away from the optimum $\inf_{\alpha \in [0,\infty)} \varepsilon_T(g_\alpha \circ \phi_\alpha)$. See the supplementary material for a proof. This optimality property is shared with related regularization settings [47, 48, 49, 42, 28].

**Two accompanying target error bounds**    Let us now discuss the reasoning above based on the two error bounds described in Section 3. First, recall the target error bound of [25] in Eq. (4). If we take $D(\alpha) := \widehat{d}_{\mathcal{G}\triangle\mathcal{G}}(\phi_\alpha(\mathbf{x}), \phi_\alpha(\mathbf{x}')) + \eta_{t,\mathcal{G},\delta}$ and $E(\alpha) := \varepsilon_S(g_\alpha \circ \phi_\alpha) + \lambda_{\mathcal{G}}(\phi_\alpha)$, it is natural to assume $D(\alpha)$ to decrease with $\alpha$, especially for adversarial approaches which penalize the empirical $\mathcal{G}\triangle\mathcal{G}$-divergence $\widehat{d}_{\mathcal{G}\triangle\mathcal{G}}(\phi(\mathbf{x}), \phi(\mathbf{x}'))$, see [16]. It also holds that $E(\alpha) \leq 3$ for all $\alpha \in [0,\infty)$. For the balancing value to exist for large sample size $t$, we need to verify that $D(\alpha)/D(0) \to 0$ for $t, \alpha \to \infty$. However, this is satisfied for most function classes $\Phi$, since it can be assumed that the constant function $\phi : \mathbf{x} \mapsto c \in \mathbb{R}$ which achieves $\widehat{d}_{\mathcal{G}\triangle\mathcal{G}}(\phi(\mathbf{x}), \phi(\mathbf{x}')) = 0$ is contained in $\Phi$. That is, $\widehat{d}_{\mathcal{G}\triangle\mathcal{G}}(\phi(\mathbf{x}), \phi(\mathbf{x}')) = 0$ can be achieved for $\alpha \to \infty$ and consequently $D(\alpha)/D(0) \to 0$ for $t, \alpha \to \infty$.

Consider now the bound in Eq. (6). By using $D(\alpha) := \widehat{d}_{h,\mathcal{H}}^{(\rho)}(\phi_\alpha(\mathbf{x}), \phi_\alpha(\mathbf{x}')) + \eta_{s+t,\mathcal{H},\delta}$, $E(\alpha) := \varepsilon_S^{(\rho)}(h \circ \phi) + \lambda_{\mathcal{H}}(\phi)$ and $B = 3$, all assumptions above are naturally satisfied similarly to the target error bound in Eq. (4) and the balancing value exists. In Section 5, we empirically investigate the performance of our method based on our two accompanying target error bounds. However, let us first show how we can overcome the problem of the non-computable term $E(\alpha)$.

### 4.2    Approximation of balancing value

Unfortunately, $E(\alpha)$ in Eq. (1) usually contains information about the target labeling function $l_T$ and $\alpha^*$ can therefore not be calculated directly.

**Balancing principle estimate**    We propose the following criterion for estimating $\alpha^*$ which is realized by the BPDA in Algorithm 1.

**Definition 2** (Balancing principle estimate). *Let $\alpha_1, \ldots, \alpha_w, w \in \mathbb{N}$ with $\alpha_1 = 0$ be an increasing sequence of values in $[0,\infty)$ and denote by $f_i := g_{\alpha_i} \circ \phi_{\alpha_i}$. Then, the balancing principle estimate is*

$$\alpha_{\mathrm{BP}} := \max\left\{\alpha_i \;\middle|\; \forall j \in \{1, \ldots, i-1\} : \widehat{\varepsilon}_T(f_i, f_j) \leq D(\alpha_j)\left(2 + \frac{2B}{D(0)}\right) + \eta_{t,\mathcal{G},\delta}\right\} \tag{12}$$

The balancing principle estimate in Definition 2 is based on checking an upper bound on the empirical cross-error $\widehat{\varepsilon}_T(f_i, f_j)$ between two models $f_i, f_j$ resulting from two different values $\alpha_i, \alpha_j$ of the

regularization parameter, respectively. The empirical cross-error does not contain information about the unknown target labels $l_T(x_1'), \ldots, l_T(x_t')$ and can be computed based on available data. The main reason why we expect the balancing principle estimate $\alpha_{\mathrm{BP}}$ to be near to the balancing value in Eq. (11) can be explained as follows.

**Lemma 1.** *Let $\delta \in (0, 1)$, $\alpha, \beta \in [0, \infty)$ and denote by $f_\alpha := g_\alpha \circ \phi_\alpha$. If $0 \le \alpha \le \beta \le \alpha^*$ then the following holds with probability at least $1 - \delta$:*

$$\widehat{\varepsilon}_T(f_\alpha, f_\beta) \le D(\alpha) \left( 2 + \frac{2B}{D(0)} \right) + \eta_{t, \mathcal{G}, \delta} \tag{13}$$

Lemma 1 (see the supplementary material for its proof) shows that the inequality in Eq. (12) is satisfied if $\alpha_j \le \alpha_i \le \alpha^*$. This implies that $\alpha_i > \alpha^*$ if the criterion is violated for some $\alpha_j \le \alpha_i$. Consequently, the maximum $\alpha_{\mathrm{BP}}$ as defined in Eq. (12) from an increasing sequence $\alpha_1, \ldots, \alpha_w$ which violates Eq. (13) for some $j \in \{1, \ldots, i - 1\}$ can be assumed to be near to $\alpha^*$.

**Generalization guarantee**  The model $g_{\alpha_{\mathrm{BP}}} \circ \phi_{\alpha_{\mathrm{BP}}}$ obtained by the BPDA in Algorithm 1 satisfies the following generalization guarantee.

**Theorem 1.** *Let $\delta \in (0, 1)$ and $\alpha_1, \ldots, \alpha_w \in [0, \infty), \alpha_1 = 0$ be an increasing sequence such that*

$$D(\alpha_l) \le q \cdot D(\alpha_{l+1}) \tag{14}$$

*for all $l \in \{1, \ldots, w - 1\}$ and some $q > 1$. Then, with probability at least $1 - \delta$*

$$\varepsilon_T(g_{\alpha_{\mathrm{BP}}} \circ \phi_{\alpha_{\mathrm{BP}}}) \le D(\alpha^*) \left( 3 + \frac{3B}{D(0)} \right) q + \eta_{t, \mathcal{G}, \delta} \tag{15}$$

Theorem 1 shows that the target error of the model $g_{\alpha_{\mathrm{BP}}} \circ \phi_{\alpha_{\mathrm{BP}}}$ identified by the BPDA has the same error rate as $D(\alpha^*)$ for $t \to \infty$. Moreover, if the optimum $\inf_{\alpha \in [0, \infty)} D(\alpha) + \overline{E}(\alpha)$ is achieved, then the error rate is optimal in the sense that $\varepsilon_T(g_{\alpha_{\mathrm{BP}}} \circ \phi_{\alpha_{\mathrm{BP}}})$ is only a constant factor worse than the minimum $\inf_{\alpha \in [0, \infty)} D(\alpha) + \overline{E}(\alpha)$. The constant factor is given by $(3 + {3B}/{D(0)}) q \max \{ {D(0)}/{B}, 1 \}$. That is, the bound is larger for steeper $D$ between two consecutive values for $\alpha$ ($q$ is larger), and, it is larger for larger ${D(0)}/{B}$. The constant can be derived by bounding $D(\alpha^*)$ in Eq. (15) as detailed in the supplementary material. If $E(\alpha)$ is increasing and the instantiation bound in Eq. (2) is tight, i.e. it holds with equality, then the error $\varepsilon_T(g_{\alpha_{\mathrm{BP}}} \circ \phi_{\alpha_{\mathrm{BP}}})$ is only a constant factor worse than the optimal error $\inf_{\alpha \in [0, \infty)} \varepsilon_T(g_\alpha \circ \phi_\alpha)$.

It is well known that appropriate assumptions as given above are needed for successful domain adaptation [52, 19]. However, in practice, target error bounds are often not tight and optimality cannot be guaranteed. In the following section, we therefore investigate the performance of our method on benchmark datasets based on our two accompanying target error bounds.

# 5  Empirical evaluations

We empirically investigate the performance of our approach based on two target error bounds, two parameter selection methods, three datasets and different domain adaptation methods[3].

## 5.1  Setup

Given a domain adaptation algorithm that follows Eq. (1), the goal is to identify the regularization parameter $\alpha$ from the sequence $0, 10^{-6}, 10^{-5}, \ldots, 10^3, 10^4$ which leads to the smallest target error of the model learned by the algorithm.

**Datasets**  We rely on one academic example which we call *Transformed Moons*. Transformed Moons shows a density ratio that is unbounded in large regions, see Figure 1. We also use the *Amazon Reviews* dataset [53]. This dataset contains text reviews from four domains: books (B), DVDs (D), electronics (E), and kitchen appliances (K). Reviews are encoded in 5000 dimensional feature vectors of bag-of-words unigrams and bigrams with binary labels: label 0 if the product is ranked by 1 to 3

---

[3]The source-code can be found at `https://github.com/Xpitfire/bpda`

stars, and label 1 if the product is ranked by 4 or 5 stars. From the four categories we obtain twelve domain adaptation tasks where each category serves once as source domain and once as target domain. We use the same data splits as previous works [54, 55, 16]. Thus, we have 2000 labeled source examples and 2000 unlabeled target examples for training, and between 3000 and 6000 examples for testing. Our third dataset is the *DomainNet–2019* dataset consisting of six different domains, each having 345 classes, and an average count of 288 images per class, i.e. around 0.6 million images [56]. However, our focus is not on large-scale domain adaptation, but rather on ranking model selection methods; hence, we propose a reduced version of the DomainNet–2019 dataset. In particular, we reduce the number of classes to five. We call our new dataset *MiniDomainNet*. See the supplementary material for the dataset statistics.

**Balancing principle for domain adaptation**    We apply the BPDA in Algorithm 1 based on the two accompanying target error bounds described in Section 3. On the two-class datasets Transformed Moons and Amazon Reviews, we apply the BPDA based on Eq. (4). We set the domain distance $D(\alpha) := \widehat{d}_{\mathcal{G}\triangle\mathcal{G}}(\phi_\alpha(\mathbf{x}), \phi_\alpha(\mathbf{x}')) + \eta_{t,\mathcal{G},\delta}$ and approximate its value by a classifier as proposed in [16, Subsection 3.2]. In our experiments on the multi-class dataset MiniDomainNet, we apply the BPDA based on Eq. (6). We define $D(\alpha) := \widehat{d}_{h,\mathcal{H}}^{(\rho)}(\phi_\alpha(\mathbf{x}), \phi_\alpha(\mathbf{x}')) + \eta_{s+t,\mathcal{H},\delta}$ and approximate its value as proposed in [20, Subsection 4.2]. In both variants of the BPDA, one for each bound, we follow the argument of [25, Subsection 7.2] to have enough unlabeled data to evaluate the bounds without considering the finite sample error term $\eta_{t,\mathcal{G},\delta}$. We repeat each domain adaptation model training several times. The if-statement in the BPDA in Algorithm 1 is considered violated, if there is a violation of the statement for at least one of the repetitions. For a fair comparison, the evaluations of IWV and DEV are also based on all repetitions. More precisely, for IWV and DEV, we choose the parameter with the lowest average importance weighted risk and lowest average DEV-risk, respectively, where the average is computed over all repetitions.

**Parameter choice baselines**    We compare our parameter choice approach to four baselines. The first baseline is training on source data only (SO). The second baseline is the best target error (TB) and it serves as a lower bound for the error. The third baseline is *importance weighted validation* [3] (IWV). We follow [23] and use *held-out* validation, i.e. we hold out a part of the training data as validation set, and we compute the importance weights based on this validation set. We also follow [57] and [23, Subsection 4.3] to estimate the importance weight by a classifier trained to separate source from target data. The classifier is tuned separately for each task and dataset such that its validation misclassification error is at most 0.05. For MiniDomainNet, we compute the importance weight based on the features of the pre-trained ResNet-18 [58]. The fourth baseline is *deep embedded validation* [23] (DEV) which solves the unbounded variance issue in IWV. Following [23], DEV is applied on the features of the neural networks.

**Domain adaptation methods**    In our experiments, we use three domain adaptation techniques. Domain-adversarial neural networks [16] (DANN), Maximum Mean Discrepancy [59, 15] (MMD) and Central Moment Discrepancy [18] (CMD). The details of all neural network architectures used, as well as the training strategy and hyperparameters are provided in the supplementary material.

**Assumptions**    To evaluate the compliance of the assumptions in Subsection 4.1 for the algorithms CMD and MMD, we estimate $E$, the domain distance $D$ and the least non-decreasing majorant $\overline{E}$ on the Transformed Moons dataset. In particular, the term $E$ is computed using the (in unsupervised domain adaptation unknown) target labels for training a classifier to minimize the minimum possible combined error in Eq. (4). We repeat the evaluations 10 times for different initializations of the domain adaptation model weights.

## 5.2    Results

**Assumptions**    The terms $E$ and $D$ for MMD on Transformed Moons are shown in Figure 2 and, for CMD, in the supplementary material. The following observation (a)–(d) can be made: (a) $E$ is bounded, the mean curves of $D(\alpha)$ tend to be non-increasing and the technical assumption $D(\alpha)/D(0) < \sup_{\alpha\in[0,\infty)} E(\alpha)/B$ is satisfied for $\alpha \geq 10^3$. That is, the mean curves tend to follow the assumptions listed in Section 4.1. (b) The mean curves $\overline{E}$ and $E$ tend to be similar. (c) The average parameters chosen by the balancing principle (see $\alpha_{\mathrm{BP}}$ in Figure 2) are the maximum values for

which the mean curves of $\overline{E}(\alpha)/B$ are smaller than the mean curves of $D(\alpha)/D(0)$. That is, the BPDA described in Algorithm 1 tends to perform as expected. (d) There is a moderate trend towards a violation of the monotonicity assumption for $D$ for CMD (see the supplementary material for the figure). However, the BPDA is (on average) robust w.r.t. this violation as it picks nearly the optimal value (see Table 2).

**Comparative study**   Average values of our empirical evaluations on Transformed Moons and Amazon Review are summarized in Table 2. For the full tables on Transformed Moons, Amazon Reviews and MiniDomainNet, see the supplementary material. Although making no assumptions on the ratio between target and source density, the BPDA outperforms related parameter choice methods (IWV, DEV) on almost all tasks on Transformed Moons and Amazon Reviews, and, obtains competitive results on MiniDomainNet.

Table 2: Average target misclassification error with best values in bold. See the supplementary material for the full results for all datasets and domain adaptation tasks.

| | Transformed Moons | | | | | Amazon Reviews | | | | |
|---|---|---|---|---|---|---|---|---|---|---|
| **Method** | SO | IWV | DEV | BPDA | TB | SO | IWV | DEV | BPDA | TB |
| MMD | 0.21 | 0.20 | 0.34 | **0.16** | 0.16 | 0.27 | 0.25 | 0.25 | **0.22** | 0.21 |
| DANN | 0.18 | 0.18 | 0.17 | **0.12** | 0.12 | 0.28 | 0.28 | 0.32 | **0.27** | 0.24 |
| CMD | 0.21 | 0.20 | **0.19** | **0.19** | 0.18 | 0.28 | 0.24 | 0.25 | **0.21** | 0.21 |
| Avg. | 0.20 | 0.19 | 0.23 | **0.16** | 0.15 | 0.28 | 0.26 | 0.27 | **0.23** | 0.22 |

# 6   Conclusion and future work

One widely-used technique for unsupervised domain adaptation is to map the data into a new feature space where the source and target data representations appear similar, and where enough information is preserved for prediction [14]. The similarity is often realized by minimizing the source error weighted by a distance measure between source and target representations. One common interpretation of this approach is to learn *domain-invariant* representations. However, there is a subtle difference between *distance-regularization* by penalizing the error minimization with a distance as above, and *distance-minimization* which results in domain-invariant representations. The latter can even lead to deteriorated performance [19]. In contrast, the interpretation as regularization problem opens up a powerful toolbox of mathematical techniques. We take up the technique of balancing stability and approximation in the regularization of ill-posed inverse problems, to tackle the problem of choosing the crucial regularization parameter in distance-regularized domain adaptation. Our approach overcomes the lack of target labels, it satisfies a generalization guarantee, and is (to the best of our knowledge) the first theoretically justified method that allows source and target distributions with disjoint supports. Finally, our approach outperforms or is on par, with the state of the art on the problem of choosing the regularization parameter, on several domain adaptation methods; applied on different datasets.

## Broader impact

Many fields, such as manufacturing, personalized medicine or analytical chemistry, have to handle problems of domain shift together with issues of data limitations. These areas can profit from our research, as it provides a principled way of choosing a justified regularization parameter of unsupervised domain adaptation algorithms. Our method provides a high level of trust by applying mathematical techniques with guarantees originally developed in the area of ill-posed inverse problems. However, in domain adaptation in general, one critical point is the bias stored in the source data (e.g. past product, previous patient data, previous spectroscope), since knowledge in this data is used to improve the learning on the unlabeled target data (e.g. new product, new patient data, new spectroscope). If such a bias is present in one of the source domains, the predictions in new target domains might always suffer from this bias. We therefore suggest to not only look at new data in new target domains but more importantly consider dataset bias in already collected labeled source data.

## Acknowledgments and Disclosure of Funding

We thank Markus Holzleitner for careful proofreading. We thank the anonymous reviewers for a constructive discussion which formed the basis for the supplementary material's section about limitations, risks and future developments. The research reported in this paper has been funded by the Federal Ministry for Climate Action, Environment, Energy, Mobility, Innovation and Technology (BMK), the Federal Ministry for Digital and Economic Affairs (BMDW), and the Province of Upper Austria in the frame of the COMET–Competence Centers for Excellent Technologies Programme and the COMET Module S3AI managed by the Austrian Research Promotion Agency FFG. The LIT AI Lab is financed by the Federal State of Upper Austria. We further acknowledge the partial support of the FFG project AutoQual-I. The ELLIS Unit Linz, the LIT AI Lab and the Institute for Machine Learning, are supported by the Federal State Upper Austria. IARAI is supported by Here Technologies. We thank the projects AI-MOTION (LIT-2018-6-YOU-212), DeepToxGen (LIT-2017-3-YOU-003), AI-SNN (LIT-2018-6-YOU-214), DeepFlood (LIT-2019-8-YOU-213), Medical Cognitive Computing Center (MC3), INCONTROL-RL (FFG-881064), PRIMAL (FFG-873979), DL for GranularFlow (FFG-871302), AIRI FG 9-N (FWF-36284, FWF-36235), ELISE (H2020-ICT-2019-3 ID: 951847), AIDD (MSCA-ITN-2020 ID: 956832). We thank Janssen Pharmaceutica (MaDeSMart, HBC.2018.2287), Audi.JKU Deep Learning Center, TGW LOGISTICS GROUP GMBH, Silicon Austria Labs (SAL), FILL Gesellschaft mbH, Anyline GmbH, Google, ZF Friedrichshafen AG, Robert Bosch GmbH, UCB Biopharma SRL, Merck Healthcare KGaA, Verbund AG, TÜV Austria, and the NVIDIA Corporation. This material is based upon work supported by the Google Cloud Research Credits program with the award GCP19980904.

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
