# Supplementary material for:
# The balancing principle for parameter choice in distance-regularized domain adaptation

Werner Zellinger[1,*]    Natalia Shepeleva[1]    Marius-Constantin Dinu[2,3]

Hamid Eghbal-zadeh[4,5]    Duc Hoan Nguyen[6]    Bernhard Nessler[2]

Sergei V. Pereverzyev[6]    Bernhard A. Moser[1]

[1]Software Competence Center Hagenberg GmbH
[2]Institute for Machine Learning, Johannes Kepler University Linz
[3]Dynatrace Research
[4]Institute of Computational Perception, Johannes Kepler University Linz
[5]LIT AI Lab, Johannes Kepler University Linz
[6]Johann Radon Institute for Computational and Applied Mathematics, Austrian
Academy of Sciences
[*]`werner.zellinger@scch.at`

## 1   Balancing value of target error bound

In this work, we assume that a target error bound is given which satisfies the form

$$\varepsilon_T(g_\alpha \circ \phi_\alpha) \leq D(\alpha) + E(\alpha) \tag{1}$$

such that (a) $\alpha \mapsto E(\alpha)$ is continuous and bounded by some constant $B > 0$, (b) $\alpha \mapsto D(\alpha) \geq 0$ is continuous, non-increasing and non-degenerate, i.e. $D(0) > 0$ and (c) $\lim_{\alpha \to \infty} \frac{D(\alpha)}{D(0)} < \sup_{\alpha \in [0,\infty)} \frac{E(\alpha)}{B}$. Under these assumptions, the balancing value $\alpha^*$ which achieves

$$\frac{D(\alpha^*)}{D(0)} = \frac{\overline{E}(\alpha^*)}{B} \tag{2}$$

exists, where $\overline{E}(\alpha)$ refers to the least non-decreasing majorant of $E(\alpha)$.

**Definition 1** (Least non-decreasing majorant [1])**.** *The least non-decreasing majorant of $E(\alpha)$ is given by $\overline{E}(\alpha) := \sup_{\beta \in [0,\alpha]} E(\beta)$.*

Interestingly, the terms $D(\alpha^*)$ and $D(\alpha^*) + \overline{E}(\alpha^*)$ evaluated at the balancing value $\alpha^*$ are only a constant factor away from the minimizer $\inf_{\alpha \in [0,\infty)} D(\alpha) + \overline{E}(\alpha)$.

**Lemma 0.** *If $\inf_{\alpha \in [0,\infty)} D(\alpha) + \overline{E}(\alpha)$ is achieved, then*

$$D(\alpha^*) \leq \max\left\{\frac{D(0)}{B}, 1\right\} \inf_{\alpha \in [0,\infty)} D(\alpha) + \overline{E}(\alpha) \tag{3}$$

$$D(\alpha^*) + \overline{E}(\alpha^*) \leq 2\max\left\{\frac{D(0)}{B}, \frac{B}{D(0)}\right\} \inf_{\alpha \in [0,\infty)} D(\alpha) + \overline{E}(\alpha) \tag{4}$$

35th Conference on Neural Information Processing Systems (NeurIPS 2021).

Lemma 0 proves, under certain assumptions, the optimality of the target error rate for the model $g_{\alpha^*} \circ \phi_{\alpha^*}$. More precisely, Eq. (4) implies that the error $\varepsilon_T(g_{\alpha^*} \circ \phi_{\alpha^*})$ is only a constant factor away from the optimum $\inf_{\alpha \in [0,\infty)} \varepsilon_T(g_\alpha \circ \phi_\alpha)$ if, it exists, $E(\alpha)$ is non-decreasing and Eq. (1) holds with equality.

*Proof.* Denote by $\alpha_{\mathrm{opt}} \in [0,\infty)$ the value achieving the infimum of $D(\alpha) + \overline{E}(\alpha)$. If $\alpha_{\mathrm{opt}} \le \alpha^*$, then the definition of $\alpha^*$ and assumption (b) imply

$$\overline{E}(\alpha^*) \frac{D(0)}{B} \le D(\alpha^*) \le D(\alpha_{\mathrm{opt}}) \le \inf_{\alpha \in [0,\infty)} D(\alpha) + \overline{E}(\alpha) \tag{5}$$

If $\alpha_{\mathrm{opt}} > \alpha^*$ then the definition of $\alpha^*$ and Definition 1 imply

$$D(\alpha^*) \frac{B}{D(0)} \le \overline{E}(\alpha^*) \le \overline{E}(\alpha_{\mathrm{opt}}) \le \inf_{\alpha \in [0,\infty)} D(\alpha) + \overline{E}(\alpha) \tag{6}$$

Combining Eq. (5) and Eq. (6) for $D(\alpha^*)$ gives

$$D(\alpha^*) \le \max\left\{ \frac{D(0)}{B}, 1 \right\} \inf_{\alpha \in [0,\infty)} D(\alpha) + \overline{E}(\alpha) \tag{7}$$

Combining Eq. (5) and Eq. (6) for $\overline{E}(\alpha^*)$ gives

$$\overline{E}(\alpha^*) \le \max\left\{ \frac{B}{D(0)}, 1 \right\} \inf_{\alpha \in [0,\infty)} D(\alpha) + \overline{E}(\alpha) \tag{8}$$

Summing Eq. (7) and Eq. (8) yields

$$\begin{aligned} D(\alpha^*) + \overline{E}(\alpha^*) &\le \max\left\{ \frac{D(0)}{B}, 1 \right\} \inf_{\alpha \in [0,\infty)} D(\alpha) + \overline{E}(\alpha) \\ &\quad + \max\left\{ \frac{B}{D(0)}, 1 \right\} \inf_{\alpha \in [0,\infty)} D(\alpha) + \overline{E}(\alpha) \\ &\le 2 \max\left\{ \frac{B}{D(0)}, \frac{D(0)}{B} \right\} \inf_{\alpha \in [0,\infty)} D(\alpha) + \overline{E}(\alpha) \end{aligned}$$

$\square$

## 2 Criterion for approximating the balancing value

Recall that we assume that the target cross-errors satisfy some concentration inequality

$$|\varepsilon_T(f,g) - \widehat{\varepsilon}_T(f,g)| \le \eta_{t,\mathcal{F},\delta} \tag{9}$$

which holds with probability at least $1-\delta$ uniformly over all $f,g \in \mathcal{F}$ for some $\eta_{t,\mathcal{F},\delta} \in \mathbb{R}$ such that $\eta_{t,\mathcal{F},\delta} \to 0$ for $t \to \infty$. The main criterion used to define the balancing principle is as follows.

**Lemma 1.** *Let $\delta \in (0,1)$, $\alpha, \beta \in [0,\infty)$ and denote by $f_\alpha := g_\alpha \circ \phi_\alpha$. If $0 \le \alpha \le \beta \le \alpha^*$ then the following holds with probability at least $1-\delta$:*

$$\widehat{\varepsilon}_T(f_\alpha, f_\beta) \le D(\alpha)\left( 2 + \frac{2B}{D(0)} \right) + \eta_{t,\mathcal{G},\delta} \tag{10}$$

*Proof of Lemma 1.* The following inequalities are all to be understood to hold with probability at least $1-\delta$. For all $\alpha \le \beta \le \alpha^*$, Eq. (9) and the triangle inequality give

$$\begin{aligned} \widehat{\varepsilon}_T(f_\alpha, f_\beta) &\le \varepsilon_T(f_\alpha, f_\beta) + \eta_{t,\mathcal{G},\delta} \\ &\le \varepsilon_T(f_\alpha) + \varepsilon_T(f_\beta) + \eta_{t,\mathcal{G},\delta} \end{aligned}$$

Using the instantiation bound of the balancing principle in Eq. (1) further implies that

$$\widehat{\varepsilon}_T(f_\alpha, f_\beta) \le D(\alpha) + E(\alpha) + D(\beta) + E(\beta) + \eta_{t,\mathcal{G},\delta}$$

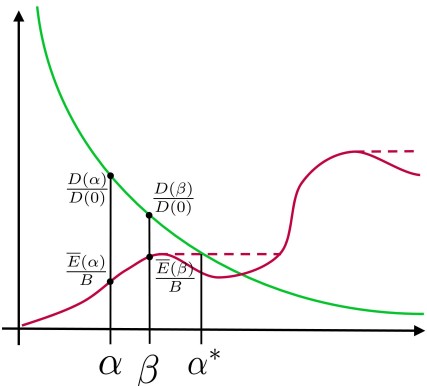

Figure 1: The proof of Lemma 1 is based on the monotonicity of $\frac{D(\alpha)}{D(0)}$ (green) and the monotonicity of the least non-decreasing majorant $\frac{\overline{E}(\alpha)}{B}$ (red dashed) of $\frac{E(\alpha)}{B}$ (red).

Definition 1 of the least non-decreasing majorant gives

$$\widehat{\varepsilon}_T(f_\alpha, f_\beta) \le D(\alpha) + \overline{E}(\alpha) + D(\beta) + \overline{E}(\beta) + \eta_{t,\mathcal{G},\delta}$$

Finally, we follow [2] and use the monotonicity of $D$ and $\overline{E}$ to obtain

$$\widehat{\varepsilon}_T(f_\alpha, f_\beta) \le D(\alpha) + \frac{BD(\alpha)}{D(0)} + D(\beta) + \frac{BD(\beta)}{D(0)} + \eta_{t,\mathcal{G},\delta}$$

$$\le D(\alpha)\left(2 + \frac{2B}{D(0)}\right) + \eta_{t,\mathcal{G},\delta}$$

Figure 1 provides a helpful illustration for the last two steps. $\qquad\square$

## 3 Generalization guarantee for balancing principle estimate

Our main theorem is stated as follows.

**Theorem 1.** *Let $\delta \in (0,1)$ and $\alpha_1, \ldots, \alpha_w \in [0, \infty), \alpha_1 = 0$ be an increasing sequence such that*

$$D(\alpha_l) \le q \cdot D(\alpha_{l+1}) \tag{11}$$

*for all $l \in \{1, \ldots, w-1\}$ and some $q > 1$. Then, with probability at least $1 - \delta$*

$$\varepsilon_T(g_{\alpha_{\mathrm{BP}}} \circ \phi_{\alpha_{\mathrm{BP}}}) \le D(\alpha^*)\left(3 + \frac{3B}{D(0)}\right)q + \eta_{t,\mathcal{G},\delta} \tag{12}$$

The following proof of Theorem 1 follows arguments from the principle of balancing stability and approximation in the theory of regularized ill-posed inverse problems. See Theorem 1 in [3] for a similar application to the adaptive choice of parameters in kernel regression.

*Proof of Thm. 1.* Let us denote by

$$\bar{\alpha} := \max\left\{\alpha_i \,\middle|\, \frac{\overline{E}(\alpha_i)}{B} \le \frac{D(\alpha_i)}{D(0)}, i \in \{1, \ldots, w\}\right\}$$

and by $f_\alpha := g_\alpha \circ \phi_\alpha$. From Eq. (10) we obtain for all $j \in \{1, \ldots, w\}$ such that $\alpha_j \le \bar{\alpha} \le \alpha^*$ with probability at least $1 - \delta$

$$\varepsilon_T(f_{\alpha_j}, f_{\bar{\alpha}}) \le D(\alpha_j)\left(2 + \frac{2B}{D(0)}\right) + \eta_{t,\mathcal{G},\delta}. \tag{13}$$

Note that $\bar{\alpha} \in \{\alpha_1, \ldots, \alpha_w\}$ and that $\alpha_{\mathrm{BP}}$ is the maximum of all $\alpha_i \in \{\alpha_1, \ldots, \alpha_w\}$ satisfying

$$\varepsilon_T(f_{\alpha_i}, f_{\alpha_j}) \le D(\alpha_j)\left(2 + \frac{2B}{D(0)}\right) + \eta_{t,\mathcal{G},\delta} \tag{14}$$

for all $j \in \{1, \ldots, i-1\}$. It follows that $\bar{\alpha} \leq \alpha_{\mathrm{BP}}$. Moreover, with probability at least $1 - \delta$,

$$
\begin{aligned}
\varepsilon_T(f_{\alpha_{\mathrm{BP}}}) &\leq \varepsilon_T(f_{\alpha_{\mathrm{BP}}}, f_{\bar{\alpha}}) + \varepsilon_T(f_{\bar{\alpha}}) \\
&\leq D(\bar{\alpha}) \left( 2 + \frac{2B}{D(0)} \right) + \eta_{t,\mathcal{G},\delta} + \varepsilon_T(f_{\bar{\alpha}}) \\
&\leq D(\bar{\alpha}) \left( 2 + \frac{2B}{D(0)} \right) + \eta_{t,\mathcal{G},\delta} + D(\bar{\alpha}) + E(\bar{\alpha}) \\
&\leq D(\bar{\alpha}) \left( 2 + \frac{2B}{D(0)} \right) + \eta_{t,\mathcal{G},\delta} + D(\bar{\alpha}) + B\frac{D(\bar{\alpha})}{D(0)} \\
&= D(\bar{\alpha}) \left( 3 + \frac{3B}{D(0)} \right) + \eta_{t,\mathcal{G},\delta}
\end{aligned}
$$

where we used the triangle inequality and Lemma 1 to prove the first two inequalities, followed by Eq. (1) and the same monotonicity argument as used in the proof of Lemma 1, see also Figure 1.

Finally, let $l$ be such that $\bar{\alpha} =: \alpha_l \leq \alpha^* \leq \alpha_{l+1}$. Since $D$ is non-increasing, we obtain $qD(\alpha^*) \geq qD(\alpha_{l+1})$ and, by assumption, $qD(\alpha_{l+1}) \geq D(\alpha_l) = D(\bar{\alpha})$. The final inequality is shown by recalling that $\bar{\alpha} \leq \alpha^*$. $\qquad\square$

## 4  MiniDomainNet dataset

The parameter selection methods for domain adaptation require to train several models with various parameters. Furthermore, evaluation of such methods include applying various domain adaptation techniques, which results in high computational demand when using large-scale datasets. In order to reduce the computational resources needed in this area, while keeping the difficulty of working with high-resolution images, and working on a problem with several domains, we fork a smaller version of the DomainNet dataset [4], which we call the *MiniDomainNet* dataset. MiniDomainNet makes research on the area of parameter selection for domain adaptation more accessible, by significantly reducing the computational needs, while providing a challenging, and sufficiently-large test bed for evaluating deep models.

The DomainNet dataset consists of approximately $0.6$ million images divided into 6 domains (Quick-draw, Real, Clipart, Sketch, Infograph, and Painting), with each domain having $345$ classes. The average count of images of DomainNet in each class, and across all domains is approx. $288$. We curate the MiniDomainNet dataset from the DomainNet dataset as follows. We select the top-five largest classes in the training set of DomainNet, based on the highest average image-count per class across all domains. This selection process will result in a dataset with the largest amount of training data per class, which is ideal for training deep models.

In our experiments with MiniDomainNet, we follow a recommendation in [4], that uses a combined-source setting. To define our domain adaptation tasks, we select $5$ out of the $6$ domains and combine them into our *combined source* dataset. We use the remaining domain as our target dataset. By permuting all source combinations, we then define 6 domain adaptation tasks, which we refer to as combined-source datasets (CS, as denoted in Table 5).

In addition to providing the MiniDomainNet, we further address an issue regarding the currently available version of the DomainNet dataset. During our development process, we found that $10$ files from the class $327$ (t-shirt) in the painting domain sub-set, were missing in the file list of the training set (`painting_train.txt`). We provide a fix for this issue in our source code, by inserting the missing class references and their corresponding files. The fix can be found in the source code, in `dataloaders/domainnet.py`.

## 5  Extended empirical evaluations

In this section, we provide details of our training setup, the computational resources used to conduct the experiments, the model selection procedures, and, our evaluation results.

## 5.1 Details for training

**Transformed Moons** On the Transformed Moons dataset, we use a feed-forward network with two fully-connected layers, with 16 nodes each, followed by ReLU non-linearity. The network is optimized by Adam [5] optimizer for 250 epochs, with $\beta_1 = 0.9$, $\beta_2 = 0.999$, and the initial learning rate of 0.01, using a MultiStep scheduler which halved the learning rate in epochs 50, 100, and 150. To train proxy-A classifiers (required in [6]), we use 1 fully-connected layer with 16 nodes; and are trained with Adam optimizer for 200 epochs, with $\beta_1 = 0.9$, $\beta_2 = 0.999$, and an initial learning rate of 0.01, and a MultiStep scheduler halving learning rate on epochs 50, 100, 150.

**Amazon Reviews** For Amazon Reviews, we follow [7] and use a feed-forward network three fully-connected layers, with 100 nodes each, and sigmoid non-linearity. The optimizer, learning rate, and scheduler are the same as in the Transformed Moons experiments (see above). We train each model for CMD and MMD experiments for 50 epochs and for DANN for 500. To estimate the $\mathcal{G}\triangle\mathcal{G}$-divergence, we follow [7] and train a classifier for separating the source sample and the target sample. In particular we apply 2 fully-connected layers with 100 nodes each and use the Adam optimizer for 200 epochs, with $\beta_1 = 0.9$, $\beta_2 = 0.999$, the initial learning rate of 0.01, and, a MultiStep scheduler halving learning rate on epochs 50, 100, 150.

**MiniDomainNet** Following the pre-trained setup from [4], we use a frozen ResNet-18 backbone model which was trained on ImageNet [8], and operate subsequent computations on the 512 dimensional extracted features. To alleviate overfitting effects on pre-computed features, we perform data augmentation on each batch and forward the images through the backbone each time. We incorporate zero padding before resizing the images to $256 \times 256$ to avoid image distortions. Following the guidance for data augmentation techniques from [9], we perform random resized cropping to $224 \times 224$ with a random viewport between $70\%$ and $100\%$ of the original image, random horizontal flipping, color jittering of $0.25\%$ on each RGB channel, and a $\pm 2$ degree rotation. After the ResNet-18 backbone output, we add several projection layers, and define the domain adaptation layers on which we use the domain adaptation methods to align the representations. The first layers are defined as a common architecture across the different domain adaptation methods. Additional layers are further added for the classification networks, according to the requirements of the individual domain adaptation methods in CMD or MMD. The number of layers/neurons in the upper layers of our architecture have been tuned in order to achieve the best performance in the source-only setup. See Table 1 for a detailed description of the architecture used. We perform experiments on all 6 domain adaptation tasks as defined in 5.4 for each of the previously listed methods. All methods have been trained for 50 epochs with Adam optimizer, an initial learning rate of 0.001, $\beta_1 = 0.9$, $\beta_2 = 0.999$, and a MultiStep learning rate scheduler, halving the learning rate after 15 and 35 epochs.

To apply the balancing principle, we require the training of an additional MDD classifier, see [10]), using the features of the adaptation layer from CMD and MMD, which is further used to calculate the MDD distance. The architecture of the MDD classifier is listed in Table 2. The MDD classifiers are trained with Adam optimizer, initial learning rate of 0.0001, and a MultiStep scheduler halving the learning rate after 15 and 25 epochs, and in total we run them for 35 epochs. For selecting the disparity parameters we followed the guidance from [11], and set $\gamma = 1.1$ in the MDD training loss, and $\rho = 0$ for calculating the MDD distance employed in BPDA.

## 5.2 Details for computational resources and source code

In experiments on Transformed Moons and Amazon Reviews, we used two HPC stations with in total 8xNVIDIA TITAN RTX 24GB, 4xIntel Xeon Scalable Processors Skylake Gold 6130 (2.10 GHz) and Ubuntu 18.04. All methods have been implemented in python using the *Pytorch library* [12]. We use *Scikit-learn library* [13] for evaluation measures and toy datasets, and the *TQDM library* [14], and *Tensorboard* [15] for keeping track of the progress of our experiments.

## 5.3 Details for model selection

**Transformed Moons and Amazon Reviews** IWV [16], DEV [11], and BPDA are used to choose the best parameter $\alpha \in \{0, 10^{-6}, 10^{-5}, \ldots, 10^3, 10^4\}$, for three different distance-regularized domain adaptation methods, namely DANN [7], MMD [17, 18] and CMD [19]. The Transformed Moons and Amazon Reviews datasets contain only 2 classes; hence, BPDA is employed using the

Table 1: Architectural listing of all layers for training on the MiniDomainNet dataset.

| | Common Architecture | |
|---|---|---|
| | **Layers** | Values |
| Backbone Output Layer | ResNet-18 (Adaptive Average Pooling Layer) | 512 |
| Projection Layers | Fully-connected Layer | 1024 |
| | Batch Normalization 1D Layer | |
| | ReLU | |
| | Fully-connected Layer | 1024 |
| | Batch Normalization 1D Layer | |
| | ReLU Activation | |
| | Dropout Layer | 0.5 |
| | Fully-connected Layer | 1024 |
| | Batch Normalization 1D Layer | |
| | ReLU Activation | |
| | Dropout Layer | 0.5 |
| | Fully-connected Layer | 1024 |
| | Batch Normalization 1D Layer | |
| | ReLU Activation | |
| | Dropout Layer | 0.5 |
| Adaptation Layers | Fully-connected Layer | 512 |
| | Batch Normalization 1D Layer | |
| | ReLU Activation | |
| | Dropout Layer | 0.5 |
| | Fully-connected Layer | 512 |
| | Batch Normalization 1D Layer | |
| | ReLU Activation | |
| | **CMD** | |
| Class Output Layer | Fully-connected Layer | 5 |
| | **MMD** | |
| Class Output Layer | Fully-connected Layer | 5 |

Table 2: MDD classifier architecture for CMD and MMD.

| | MDD Classifier | |
|---|---|---|
| | **Layers** | Values |
| Backbone Output Layer | CMD/MMD-Method Adaptation Layer | 512 |
| Projection Layers | Fully-connected Layer | 512 |
| | Batch Normalization 1D Layer | |
| | ReLU | |
| Class Output Layer | Fully-connected Layer | 5 |

bound introduced in [6]. All the methods (DANN, MMD and CMD) are repeated 10 times for each parameter $\alpha$. The if-statement in the BPDA in Algorithm 1 is considered violated, if there is a violation of the statement for at least one of the repetitions. For a fair comparison, the evaluations of IWV and DEV are also based on 10 repetitions. More precisely, for IWV and DEV, we choose the parameter with the lowest average importance weighted risk and lowest average DEV-risk, respectively, where the average is computed over all 10 repetitions.

**MiniDomainNet** IWV, DEV, and BPDA are used to choose the best parameter $\alpha \in \{0, 10^{-3}, 10^{-2}, 10^{-1}, 1, 10\}$, for two different distance-regularized domain adaptation methods, namely MMD, and CMD. Since the MiniDomainNet dataset contains 5 classes, we use the bound proposed in [10] to instantiate the BPDA. The training procedure is the same as for Transformed Moons and Amazon Reviews.

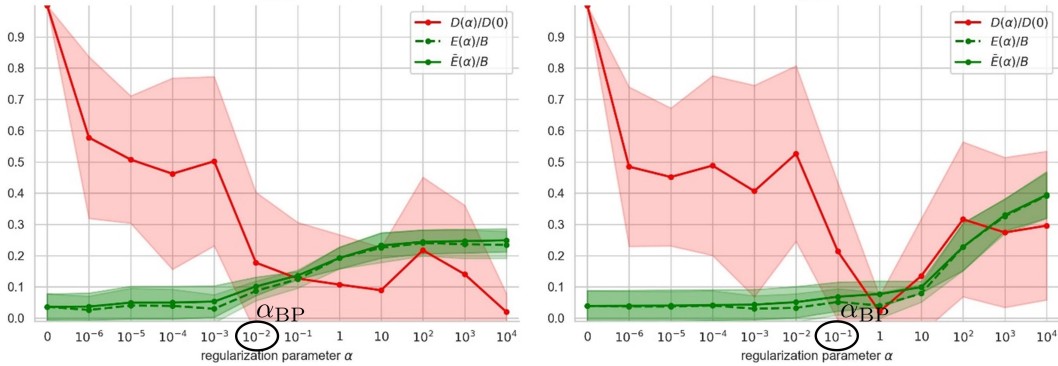

Figure 2: Average and standard deviation over 10 repetitions of estimated learning errors $E$ (in unsupervised domain adaptation unknown) and the domain distance $D$ of the accompanying target error bound [6] for distance-regularized domain adaptation models with the Maximum Mean Discrepancy [20] (left) and the Central Moment Discrepancy (right).

## 5.4 Results

This section provides empirical evidence for the compliance of the empirical settings with our assumptions made in Section 4.1 of the main document, and, gives results on the three aforementioned datasets, comparing our approach with the state of the art in parameter selection for domain adaptation.

**Verification of assumptions** Figure 2 shows the behaviour of the curves $D(\alpha)/D(0)$, $E(\alpha)/B$ and $\overline{E}(\alpha)/B$ for the two methods CMD and MMD and the Transformed Moons dataset. The following observations can be made.

- $E$ is bounded, the mean curves of $D(\alpha)$ tend to be non-increasing and the technical assumption $D(\alpha)/D(0) < \sup_{\alpha \in [0,\infty)} E(\alpha)/B$ is satisfied for $\alpha \geq 10^3$. That is, the mean curves tend to follow the assumptions.

- The mean curves $\overline{E}$ and $E$ tend to be similar. That is, the risk which is described in the main document when considering label shift, does not apply.

- The average parameters chosen by the balancing principle (see $\alpha_{\mathrm{BP}}$ in Figure) are the maximum values for which the mean curves of $E(\alpha)/B$ are smaller than the mean curves of $D(\alpha)/D(0)$. That is, the BPDA described in Algorithm 1 tends to perform as expected.

- There is a small trend towards a violation of the monotonicity assumption for $D$ in the right sub-figure for CMD. However, the BPDA is (on average) robust w.r.t. this violation as it picks nearly the optimal value. The corresponding numbers can be found in Table 2.

**Transformed Moons** The results are provided in Table 3. It can be observed that BPDA achieves the lowest average classification error among all methods, over all domain adaptation techniques.

Table 3: Average target classification error (and standard deviation) for different regularization parameter choices on the Transformed Moons dataset. 10 repetitions with different random initialization of model weights are used to estimate the importance weighted risk, the DEV risk and the BPDA. The BPDA is computed using the bound in [6].

| Method | SO | IWV | DEV | BPDA | TB |
|---|---|---|---|---|---|
| MMD | 0.205 ($\pm$0.025) | 0.199 ($\pm$0.031) | 0.339 ($\pm$0.065) | **0.157** ($\pm$**0.069**) | 0.157 ($\pm$0.069) |
| DANN | 0.177 ($\pm$0.032) | 0.177 ($\pm$0.032) | 0.169 ($\pm$0.075) | **0.115** ($\pm$**0.098**) | 0.115 ($\pm$0.098) |
| CMD | 0.205 ($\pm$0.026) | 0.198 ($\pm$0.022) | 0.190 ($\pm$0.051) | **0.185** ($\pm$**0.039**) | 0.181 ($\pm$0.038) |
| Avg. | 0.196 ($\pm$0.028) | 0.191 ($\pm$0.028) | 0.232 ($\pm$0.064) | **0.152** ($\pm$**0.069**) | 0.151 ($\pm$0.068) |

**Amazon Reviews** Table 4 shows the results of three model selection methods IWV, DEV, and BPDA which are used to choose the best parameter $\alpha$ in the sequence of $0, 10^{-6}, 10^{-5}, \ldots, 10^3, 10^4$,

for three different distance-regularized domain adaptation methods, namely DANN, MMD, and CMD. The 4 domains contained in Amazon Reviews are denoted in the tables as: books (B), DVDs (D), electronics (E), and kitchen appliances (K). As can be seen, our method achieves the lowest averaged classification error across all tasks using the MMD method. These results are consistent across all domain adaptation techniques.

**MiniDomainNet**  Table 5 shows the results of the three model selection methods IWV, DEV, and BPDA. We omitted the experiments with DANN on the MiniDomainNet dataset due to our computational limits. The 6 domains in the MiniDomainNet are denoted in the tables as: Quickdraw (Q), Real (R), Sketch (S), Clipart (C), Infograph (I), and Painting (P). Since the source domain is always a combination of all the other domains except the target, we refer to the source as Combined Source (CS). As can be seen, our method achieves the lowest averaged classification error across all tasks using the CMD method. When using MMD, the three parameter selection methods perform very similar on average, with BPDA and DEV achieving the lowest average error across all tasks.

# 6  Discussion of risks and limitations

A constructive discussion with anonymous reviewers resulted in the following list of risks and limitations of the proposed BPDA method.

**Label shift**  In this work, we do not assume a unique labeling function $l_S = l_T$ for source and target domain (covariate shift assumption), but $l_S$ and $l_T$ should be similar. In fact, even if the labeling functions are different $l_S \neq l_T$, the balancing value $\alpha^*$ can be well estimated by the balancing principle estimate $\alpha_{\mathrm{BP}}$ as proven by Lemma 1. However, the quality of the balancing value $\alpha^*$ itself can be negatively affected by excessive label shift. For example, the function $E$ can first increase strongly and then decrease (caused by label shift) which implies an increasing difference between $\overline{E}(\alpha)$ and $E(\alpha)$ for increasing $\alpha$. In such situations, the target error of the model $g_{\alpha^*} \circ \phi_{\alpha^*}$ can be high and consequently also the one of the model $g_{\alpha_{\mathrm{BP}}} \circ \phi_{\alpha_{\mathrm{BP}}}$ identified by the BPDA. However, note that our experiments (see also Figure 2) indicate similar values for $E$ and $\overline{E}$.

**Loose instantiation bound**  Our model is agnostic w.r.t. the property that different target error bounds can be chosen as a basis. However, it is known in inverse problem literature [21, 3] that a loose bound can lead to a low performance of the balancing principle which also holds for the BPDA. This problem can be approached by choosing target error bounds that take into account the specific domain shift situation, e.g. the bound [22] is suitable for general domain shift scenarios.

**Low performance of all models**  Situations exist which hinder distance-regularized domain adaptation methods to perform well. One such situation is excessive label shift as discussed above, see also [23] and references therein. Such scenarios can cause all models $f_1, \ldots, f_w$ to be inaccurate or unstable. The BPDA will select a model $f_i \in \{f_1, \ldots, f_n\}$ with low target error $\varepsilon_T(f_i)$ compared to other models in the set $\{f_1, \ldots, f_n\}$. Nevertheless, in such situations, $f_i$ might have a high target error $\varepsilon_T(f_i)$ or it might be unstable. As a solution, distance-regularized domain adaptation methods can be applied with modifications, see e.g. [22, 24, 25, 26].

**Focus on weight parameter**  Our theoretical guarantees and the high empirical performance come at the cost of focusing the selection process on the distance-penalizing parameter $\alpha$. This is in contrast to other model selection methods, such as [16, 11, 27, 28, 29], which can select different types of parameters. This limitation can be approached by employing ideas from multipenalty regularization of inverse problems to combine the BPDA with related approaches [30]. One approach is to explore a grid of admissible values of several parameters by applying the balancing principle with respect to one of them and allowing others to take all corresponding grid values. As the result of such an application, one obtains a much reduced set of parameter combinations of interest and one can apply the balancing principle or related approaches w.r.t. the other parameters. See e.g. [31] for a recent application of this approach.

## Acknowledgments and Disclosure of Funding

We thank Markus Holzleitner for careful proofreading. We thank the anonymous reviewers for a constructive discussion which formed the basis for the supplementary material's section about limitations, risks and future developments. The research reported in this paper has been funded by the Federal Ministry for Climate Action, Environment, Energy, Mobility, Innovation and Technology (BMK), the Federal Ministry for Digital and Economic Affairs (BMDW), and the Province of Upper Austria in the frame of the COMET–Competence Centers for Excellent Technologies Programme and the COMET Module S3AI managed by the Austrian Research Promotion Agency FFG. The LIT AI Lab is financed by the Federal State of Upper Austria. We further acknowledge the partial support of the FFG project AutoQual-I. The ELLIS Unit Linz, the LIT AI Lab and the Institute for Machine Learning, are supported by the Federal State Upper Austria. IARAI is supported by Here Technologies. We thank the projects AI-MOTION (LIT-2018-6-YOU-212), DeepToxGen (LIT-2017-3-YOU-003), AI-SNN (LIT-2018-6-YOU-214), DeepFlood (LIT-2019-8-YOU-213), Medical Cognitive Computing Center (MC3), INCONTROL-RL (FFG-881064), PRIMAL (FFG-873979), S3AI (FFG-872172), DL for GranularFlow (FFG-871302), AIRI FG 9-N (FWF-36284, FWF-36235), ELISE (H2020-ICT-2019-3 ID: 951847), AIDD (MSCA-ITN-2020 ID: 956832). We thank Janssen Pharmaceutica (MaDeSMart, HBC.2018.2287), Audi.JKU Deep Learning Center, TGW LOGISTICS GROUP GMBH, Silicon Austria Labs (SAL), FILL Gesellschaft mbH, Anyline GmbH, Google, ZF Friedrichshafen AG, Robert Bosch GmbH, UCB Biopharma SRL, Merck Healthcare KGaA, Verbund AG, TÜV Austria, and the NVIDIA Corporation. This material is based upon work supported by the Google Cloud Research Credits program with the award GCP19980904.

Table 4: Average target classification error (and standard deviation) for different regularization parameter choices on the Amazon Reviews dataset. 10 repetitions with different random initialization of model weights are used to estimate the importance weighted risk, the DEV risk and the BPDA. The BPDA is computed using the bound in [6].

| | MMD | | | | |
|---|---|---|---|---|---|
| **Task** | SO | IWV | DEV | BPDA | TB |
| B→D | 0.225 (±0.004) | **0.190** (**±0.004**) | 0.211 (±0.005) | **0.190** (**±0.004**) | 0.190 (±0.004) |
| B→E | 0.307 (±0.010) | 0.307 (±0.010) | **0.211** (**±0.005**) | 0.221 (±0.008) | 0.206 (±0.012) |
| B→K | 0.266 (±0.004) | **0.185** (**±0.009**) | 0.266 (±0.004) | **0.185** (**±0.009**) | 0.185 (±0.009) |
| D→B | 0.278 (±0.008) | **0.240** (**±0.007**) | 0.268 (±0.006) | 0.243 (±0.006) | 0.230 (±0.007) |
| D→E | 0.273 (±0.004) | 0.273 (±0.007) | 0.249 (±0.007) | **0.207** (**±0.004**) | 0.189 (±0.008) |
| D→K | 0.266 (±0.004) | 0.266 (±0.004) | **0.197** (**±0.006**) | **0.197** (**±0.006**) | 0.187 (±0.007) |
| E→B | 0.306 (±0.003) | 0.306 (±0.003) | 0.310 (±0.007) | **0.295** (**±0.013**) | 0.282 (±0.014) |
| E→D | 0.307 (±0.007) | 0.285 (±0.006) | 0.288 (±0.009) | **0.264** (**±0.016**) | 0.255 (±0.020) |
| E→K | 0.162 (±0.004) | **0.145** (**±0.003**) | 0.193 (±0.004) | **0.145** (**±0.003**) | 0.145 (±0.003) |
| K→B | 0.337 (±0.007) | 0.337 (±0.007) | 0.334 (±0.006) | **0.290** (**±0.010**) | 0.261 (±0.010) |
| K→D | 0.293 (±0.005) | 0.294 (±0.007) | 0.306 (±0.007) | **0.268** (**±0.010**) | 0.235 (±0.014) |
| K→E | 0.167 (±0.002) | 0.169 (±0.004) | **0.167** (**±0.002**) | 0.169 (±0.004) | 0.145 (±0.002) |
| Avg. | 0.266 (±0.005) | 0.249 (±0.005) | 0.250 (±0.005) | **0.223** (**±0.008**) | 0.209 (±0.009) |

| | DANN | | | | |
|---|---|---|---|---|---|
| **Task** | SO | IWV | DEV | BPDA | TB |
| B→D | 0.228 (±0.003) | **0.220** (**±0.011**) | 0.509 (±0.001) | 0.233 (±0.053) | 0.220 (±0.011) |
| B→E | 0.322 (±0.009) | 0.327 (±0.007) | 0.498 (±0.000) | **0.313** (**±0.081**) | 0.235 (±0.017) |
| B→K | 0.276 (±0.003) | 0.296 (±0.010) | 0.272 (±0.013) | **0.247** (**±0.103**) | 0.219 (±0.023) |
| D→B | 0.290 (±0.006) | 0.290 (±0.009) | **0.253** (**±0.045**) | **0.253** (**±0.045**) | 0.245 (±0.008) |
| D→E | 0.284 (±0.004) | 0.274 (±0.003) | 0.299 (±0.007) | **0.252** (**±0.084**) | 0.221 (±0.013) |
| D→K | 0.270 (±0.004) | 0.300 (±0.006) | 0.303 (±0.007) | **0.217** (**±0.008**) | 0.217 (±0.008) |
| E→B | 0.312 (±0.005) | **0.310** (**±0.005**) | 0.312 (±0.005) | 0.372 (±0.056) | 0.310 (±0.005) |
| E→D | 0.317 (±0.007) | **0.313** (**±0.006**) | **0.313** (**±0.006**) | 0.327 (±0.075) | 0.277 (±0.031) |
| E→K | 0.170 (±0.004) | **0.170** (**±0.004**) | **0.170** (**±0.004**) | 0.172 (±0.011) | 0.170 (±0.004) |
| K→B | 0.345 (±0.006) | 0.337 (±0.006) | 0.338 (±0.017) | **0.314** (**±0.053**) | 0.314 (±0.053) |
| K→D | 0.313 (±0.003) | 0.360 (±0.005) | 0.360 (±0.005) | **0.298** (**±0.053**) | 0.296 (±0.019) |
| K→E | 0.174 (±0.002) | **0.183** (**±0.003**) | 0.221 (±0.004) | 0.194 (±0.057) | 0.172 (±0.010) |
| Avg. | 0.275 (±0.005) | 0.282 (±0.006) | 0.321 (±0.010) | **0.266** (**±0.057**) | 0.241 (±0.017) |

| | CMD | | | | |
|---|---|---|---|---|---|
| **Task** | SO | IWV | DEV | BPDA | TB |
| B→D | 0.230 (±0.011) | **0.193** (**±0.006**) | 0.231 (±0.007) | **0.193** (**±0.006**) | 0.193 (±0.006) |
| B→E | 0.319 (±0.013) | 0.309 (±0.010) | 0.308 (±0.009) | **0.218** (**±0.011**) | 0.218 (±0.011) |
| B→K | 0.269 (±0.005) | 0.230 (±0.007) | 0.269 (±0.006) | **0.186** (**±0.010**) | 0.186 (±0.010) |
| D→B | 0.290 (±0.015) | 0.258 (±0.009) | 0.245 (±0.008) | **0.228** (**±0.006**) | 0.228 (±0.006) |
| D→E | 0.280 (±0.009) | 0.267 (±0.007) | 0.280 (±0.006) | **0.203** (**±0.007**) | 0.203 (±0.007) |
| D→K | 0.264 (±0.004) | **0.194** (**±0.006**) | 0.194 (±0.006) | **0.194** (**±0.006**) | 0.194 (±0.006) |
| E→B | 0.314 (±0.009) | 0.307 (±0.006) | 0.302 (±0.005) | **0.279** (**±0.010**) | 0.279 (±0.010) |
| E→D | 0.320 (±0.020) | 0.287 (±0.006) | 0.287 (±0.006) | **0.258** (**±0.014**) | 0.258 (±0.014) |
| E→K | 0.174 (±0.013) | 0.152 (±0.005) | 0.169 (±0.006) | **0.139** (**±0.005**) | 0.139 (±0.005) |
| K→B | 0.346 (±0.022) | **0.264** (**±0.007**) | 0.331 (±0.006) | **0.264** (**±0.007**) | 0.264 (±0.007) |
| K→D | 0.314 (±0.013) | **0.248** (**±0.006**) | 0.248 (±0.006) | **0.248** (**±0.006**) | 0.248 (±0.006) |
| K→E | 0.178 (±0.007) | **0.147** (**±0.004**) | 0.178 (±0.007) | **0.147** (**±0.004**) | 0.147 (±0.004) |
| Avg. | 0.275 (±0.012) | 0.238 (±0.007) | 0.254 (±0.007) | **0.213** (**±0.008**) | 0.213 (±0.008) |

Table 5: Average target classification error (and standard deviation) for different regularization parameter choices on the Transformed Moons dataset. 3 repetitions with different random initialization of model weights are used to estimate the importance weighted risk, the DEV risk and the BPDA. The BPDA is computed using the bound in [10].

| | MMD | | | | |
|---|---|---|---|---|---|
| **Task** | SO | IWV | DEV | BPDA | TB |
| CS→Q | 0.568(±0.007) | **0.629(±0.023)** | **0.629(±0.023)** | **0.629(±0.023)** | 0.568(±0.007) |
| CS→R | 0.068(±0.009) | 0.102(±0.020) | 0.102(±0.020) | **0.098(±0.006)** | 0.068(±0.009) |
| CS→S | 0.309(±0.010) | 0.307(±0.001) | 0.324(±0.021) | **0.296(±0.012)** | 0.296(±0.012) |
| CS→C | 0.246(±0.016) | **0.264(±0.013)** | **0.264(±0.013)** | **0.264(±0.013)** | 0.246(±0.016) |
| CS→I | 0.605(±0.012) | 0.577(±0.004) | **0.564(±0.001)** | 0.589(±0.021) | 0.564(±0.001) |
| CS→P | 0.178(±0.006) | **0.212(±0.012)** | 0.202(±0.011) | 0.214(±0.007) | 0.178(±0.006) |
| Avg. | 0.329(±0.010) | 0.349(±0.012) | 0.348(±0.015) | **0.348(±0.014)** | 0.320(±0.009) |

| | CMD | | | | |
|---|---|---|---|---|---|
| **Task** | SO | IWV | DEV | BPDA | TB |
| CS→Q | 0.568(±0.007) | 0.568(±0.007) | 0.812(±0.000) | **0.410(±0.008)** | 0.410(±0.008) |
| CS→R | 0.068(±0.009) | **0.068(±0.009)** | 0.841(±0.000) | 0.100(±0.010) | 0.068(±0.009) |
| CS→S | 0.309(±0.010) | 0.305(±0.012) | 0.875(±0.000) | **0.298(±0.005)** | 0.298(±0.005) |
| CS→C | 0.246(±0.016) | **0.257(±0.023)** | 0.850(±0.000) | 0.282(±0.032) | 0.246(±0.016) |
| CS→I | 0.605(±0.012) | **0.556(±0.031)** | 0.883(±0.000) | 0.601(±0.016) | 0.556(±0.031) |
| CS→P | 0.178(±0.006) | **0.246(±0.022)** | 0.986(±0.000) | 0.293(±0.019) | 0.178(±0.006) |
| Avg. | 0.329(±0.010) | 0.333(±0.017) | 0.875(±0.000) | **0.331(±0.015)** | 0.293(±0.013) |