# OpenReview forum: "The balancing principle for parameter choice in distance-regularized domain adaptation"
_NeurIPS.cc/2021/Conference — NeurIPS 2021 Poster_

### Official Review · Reviewer_mWtP · 2021-06-25

**Rating:** 6
**Confidence:** 4

**Summary:**

In this paper, the authors study the hyperparameter selection problem in the context of unsupervised domain adaptation.
While hyperparameter selection is not trivial at all due to the absence of the labeled data in the target domain, the authors employ the balancing principle to draw a connection between unsupervised domain adaptation and ill-posed inverse problems.
The proposed method is based on balancing the discrepancy term and the error term, and the joint minimizer is chosen as a good hyperparameter.
Notably, this method does not require any assumptions such as covariate shift and support matching as supposed in the existing researches.

**Limitations And Societal Impact:**

The authors have adequately mentioned potential social impacts in the concluding section.

**Main Review:**

I think the problem studied in this paper is pretty interesting.
Hyperparameter selection is one of the central obstacles when unsupervised domain adaptation (UDA) is applied in real-world scenarios.
While this problem itself has been actively studied as a matter of course, the proposed approach sheds new light on the estimation methodology to deal with "inaccessible" in the common domain adaptation bounds---the joint error minimizer.
This obstacle was mitigated by the balancing principle estimate (Definition 2), and in particular, by means of Lemma 1.
In my opinion, this is a pretty interesting theoretical result in the UDA context.

I'm overall positive about this paper, but let me put a few remarks that could help the authors to improve the paper.

**Clarity issue**:
I would like to see the following points improved.
- The definitions of the discrepancy and error terms $D(\alpha), E(\alpha)$ is ambiguous as of Section 2, which may lead to some potential misunderstanding. For example, "... cannot be estimated similarly to the learning errors $E(\alpha)$" (l.187), but I thought the error term only depends on the source domain, which turned out to be false at l.203 and I understood that $E(\alpha)$ contains the joint error $\lambda_\mathcal{G}$ as well.
- Why do we put the balancing condition as in Eq. (11), not $D(\alpha^*) = E(\alpha^*)$ as in the case of inverse problems?
- (l.136) "Using Eq. (3), the source error can be further upper bounded by the empirical source error": This is not clear because Eq. (3) is a concentration bound for the target domain. Do you assume the similar bound for the source domain, too?

**About the domain adaptation bound**:
In Eq. (4), the common domain adaptation is shown.
As far as I understand (please let me know if the following is wrong), the DA bound based on $\mathcal{G}$-divergence is not correct, although it is presented in [7].
Indeed, the proof of Theorem 1 in [7] upper bounds $\mathcal{H}\Delta\mathcal{H}$-divergence by $\mathcal{H}$-divergence, which does not hold in general.
To make it precise, we need to bound by $\mathcal{G}\Delta\mathcal{G}$-divergence as done in Theorem 2 in [27].

----
A few minor comments.
- (l.127) "the sampling error by $\eta_{t,\mathcal{G},\delta}$" instead of "the sampling error by $\eta_{z,\mathcal{G},\delta}$"?
- (Eq.(6)) the argument in $\varepsilon_S^{(\rho)}$ should be $g_h \circ \phi$?
- (l.232) The numbering of Lemma is not aligned between the main text and the supplementary material. Could you align or mention it at least?

**Time Spent Reviewing:**

three hours

---

> ### Author Response · Authors · 2021-08-09
> **Response to reviewer mWtP**
>
> We **thank you** very much for all your efforts, your careful reading of mathematical formulas and your time invested to review our work! We also thank you for acknowledging that our approach **sheds new light** on a common phenomenon and that you are **overall positive** about our paper. Our detailed comments are as follows:
> - Clarity regarding definition of $D$ and $E$: We thank you for mentioning this confusion. **We will give clearer explanations** for $D$ and $E$ earlier in the final paper together with pointers to concrete examples how $D$ and $E$ can be realized.
> - Direct application of balancing principle: Because a general distance $D$ is not necessarily larger than $\bar E$ and the balancing value $\alpha^*$ which satisfies $D(\alpha^*)=\bar E(\alpha^*)$ **does not necessarily exist**. However, $D(\alpha)/D(0)=1$ for $\alpha=0$ and $\bar E(\alpha)/B\leq 1$ for $\alpha=0$. This implies, together with the assumption (c) $\lim_{\alpha\to\infty} \frac{D(\alpha)}{D(0)}\leq \sup_{\alpha\in[0,\infty)}\frac{\bar E(\alpha)}{B}$, the existence of $\alpha^*$ such that $D(\alpha^*)/D(0)=\bar E(\alpha^*)/B$. We will add this explanation more explicit to the paper.
> - Concentration inequality for source error: We thank you very much for your careful reading. You identified an important **missing notice** in our paper. Indeed, also a concentration inequality for the source error is needed. We will change the corresponding sentence to: “If a sample error bound for the source error $\varepsilon_S$ analogously to that of target error in Eq. (3) is given, then...”
> - Domain adaptation bound of [7] in Eq. (3): **We agree**, the $\mathcal{G}$-divergence should be replaced by the $d_{\mathcal{G}\Delta\mathcal{G}}$-divergence. We thank you very much for your careful reading! We will do the correction in the paper. Please note that this has no further consequences.
> - **We will perform all suggested minor changes** except for Eq. (6) where $\varepsilon_S^{(\rho)}$ is defined as in [19] to take scoring functions as input.
>
> We **thank you** again very much for your constructive comments, help, and the time you invested for reviewing! If you are satisfied with our answers and changes, we kindly ask you to take this into account in your final decision.

---

> > ### Comment · Reviewer_mWtP · 2021-08-20
> > **Re: Response to reviewer mWtP**
> >
> > Thank you for providing detailed feedback. Now my questions are fully addressed. In the revision, the discussion on $D(\alpha) = \bar E(\alpha)$ is particularly important to be addressed.

---

> > > ### Author Response · Authors · 2021-08-20
> > > **Response to reviewer mWtP**
> > >
> > > Thank you for your kind answer. We will include the discussion about $E(\alpha)=D(\alpha)$ in our paper. We are glad that our responses could clarify all your concerns. Please take our consensus into account when reconsidering your score.

---

### Official Review · Reviewer_DYjR · 2021-07-15

**Rating:** 6
**Confidence:** 4

**Summary:**

This paper studies the problem of choosing a justified regularization parameter for unsupervised domain adaptation.
The authors make a link between ill-posed inverse problems and approaches based on a weighted combination of the source error and a distance between source and target feature representations.
They propose a framework based on balancing approximations and sampling errors to obtain a target error bound that takes into account a balance between learning errors domain distance.
This leads to develop a founded rule for the choice of the regularization parameter which can be applied with source and target distributions with disjoint supports.
A bound on the target risk is provided.
An experimental evaluation of the approach is provided using the Moons toy dataset and the Amazon reviews and DomainNet–2019 benchmark.

**Limitations And Societal Impact:**

The work is methodological and fundamental, so there is no direct negative societal impact.
The authors propose a broader impact discussion at the end of the paper.

However, in my opinion, the paper did not discuss properly the limitations of the work. The authors mention that this is discussed in Section 4.2 but as far as I can see, the only point is mentioning that the bound may not be tight and optimality cannot be guaranteed, but no precise limitations are discussed.


**Main Review:**

Originality
---------------
This paper studies the tuning of a particular hyperparameter in the context of unsupervised domain adaptation. An important strong point is the procedure is founded theoretically which is not often the case of existing methods.

Quality
-----------
The proposed method is justified theoretically and argumented.

The experimental evaluation is sound but limited in some points: other baselines and a study on how far the assumptions are verified could be added.

I would have also appreciated a thorough discussion on the target risk bounds that can be obtained from Thm1 for different divergence measures, both from theoretical and practical sides. One can wonder if the bound will not be large in some cases to match the required constraints?


Clarity
---------
The paper is clear and easy to read. The contribution is well presented.


Significance
-----------------
The problem studied is of importance in the context of domain adaptation problems.
The method has the advantage to be simple which makes it rather accessible in practice while being justified in practice.
This is the first (or at least among the first) for which the theoretical justification helds for  distributions without disjoint support.
The results are rathe good.

However the method is specialized for only one hyperparameter while many domain adaptation algorithms optimized losses with multiple objective and hyperparameters. So the contribution has a strong limitation from this point.
Another one is probably the computation cost that requires quadratic pairwise computation on the parameter range.
About the theoretical guarantee, one the one hand this is strong point to have such a guarantee, but the bound can be loose at some point. In representation spaces where the two distributions are close, the bounds seems to be infinite, which is not what one would expect here.
Experimental evaluation can be improved in some points




Comments
---------------------

-In order to assess the quality of the result, I would be interested in having more discussions about the bound provided in Thm 1.
It could be interesting to try to instantiate the bound in theory and in practice to make it more concrete and maybe to better compare it to other bounds.

One thing that puzzles me is that when considering a representation space $\phi$ such that the source and target distribution are naturally close, then the divergence will be naturally small and thus D(0), which makes the bound very loose and possibly infinite if the two distributions are the same.
Maybe refining the bound with restrictions on $\phi$ could be a direction for trying to have a better consistency.

-About the assumptions (paragraph started line 174): The authors have justified that they can be reasonable in theory, but I wonder what happens in practice. It could be interesting to check in the experiments to which the assumptions are verified and if the worst results can be identified as results obtained when the assumptions are not (fully) satisfied.

-In the results, and in particular in those related in the supplementary material, it seems that the proposed method has on average a higher variance than other methods. Can the authors provide some explanations for explaining this behavior. Is it a consequence of a generalization bound not that tight?
While the method is simple, it may then be more difficult to use it in practice?

-I am wondering if the authors could discuss more the computational cost of the method that requires to compute pairwise comparisons between candidate hypothesis/models. This seems costly and one can wonder if the benefits worth the computational time cost. In particular when a user is looking for a precise tuning, the cost is prohibitive and this is confirmed in the reported experiments in supplementary material (DANN on the MiniDomainNet benchmark)
Maybe can we imagine some stochastic methods to accelerate the computation of the criterion?

-About distribution with disjoint supports, the result is good but in the experimental setup other baselines adapted to disjoint supports could be considered, (e.g. the TrCV method below).

Other (heuristic) validation methods that can be discussed in the related work and used as other baselines in the experiments based a notion of "reverse validation":
 *Erheng Zhong, Wei Fan, Qiang Yang, Olivier Verscheure, Jiangtao Ren: Cross Validation Framework to Choose amongst Models and Datasets for Transfer Learning. ECML/PKDD 2010
 *Lorenzo Bruzzone, Mattia Marconcini: Domain Adaptation Problems: A DASVM Classification Technique and a Circular Validation Strategy. IEEE Trans. Pattern Anal. Mach. Intell. 32(5): 770-787 (2010)

Another direction can also be to consider heterogeneous domain adaptation, but here specific domain distances must be considered, as well as benchmarks dedicated to these tasks.

-In the experiments, I did not understand well how the baseline TB corresponding to the best target error is computed?
It is the best target error found among the $\alpha$ values selected or is it a bound obtained when learning for labeled target data?
In both cases, this is clearly a lower bound for the error.
In the first case, one can notice from the results in Appendix, that sometimes the method is close to this bound and sometimes not. In the second situation does this mean the assumption that D(\alpha) is non-increasing is false in practice.
In the second case, it is strange to be sometimes very close to the TB results.
Can the authors precise this point.

-In terms of writing, I agree that the inverse problem setting should be cited and discussed, but I think that the discussion on it can be reduced since at the end the two problems considered are rather different.

-A minor remark, but line.109, the notation $\phi(x)=\phi(x')$  is maybe badly chosen


-------
After Rebuttal
-------
The authors did a strong work to answer the points mentioned in the review. I appreciated the answers. I increase my score to 6.

**Time Spent Reviewing:**

8

---

> ### Author Response · Authors · 2021-08-09
> **Response to reviewer DYjR**
>
> We **thank you very much** for all the efforts, your time and constructive comments you provide for our work! We thank you also for acknowledging that the results are good, our procedure is founded theoretically **which is not often the case of existing methods** and that the method is simple which makes it **accessible in practice**. We understand that your major concerns are regarding (a) focus on only one hyperparameter, (b) computation costs, (c) tightness of bound, (d) visualizations of $D$ and $E$ in practice and (e) discussion of related approaches. In summary, **we agree** that a treatment of (a), (d) and (e) will improve our work (concrete comments and steps follow) but **we disagree** with some arguments for limited significance because of (b) computational costs and (c) bound looseness due to division by $D(0)$. Our detailed answers are as follows:
> - Focus on weighting parameter: **We agree** that the focus on only one hyperparameter is a limitation compared to other approaches. We will discuss this limitation and an extension for more parameters in our **new Limitations section** (see general comment above, Limitation 1). However, we note that this limitation in generality (w.r.t. different types of hyperparameters) **enables us** to propose the (to the best of our knowledge) first theoretically justified parameter choice rule for unsupervised domain adaptation that allows disjoint supports.
> - Tightness of bound: **We agree** that a short discussion about tightness of Theorem 1 is a good way for enhancing our work. Our plans are described in detail in the general comment above (see (a) and (b) Risk 1 and Risk 2). Our plans include a **new Limitations section**.
> - $D(0)$ implies looseness of bound: **We disagree** with the reasoning about looseness of the bound in Theorem 1. Indeed, the bound in Theorem 1 is not necessarily large if $D(0)$ is small. In fact, $D(0)$ is multiplied by $D(\alpha^*)$ which is assumed to be smaller or equal to $D(0)$. Consequently, the upper bound is smaller or equal to $3 D(\alpha^*) q+3 B q + \eta_{t,\mathcal{G},\delta}$ which **does not increase** with decreasing $D(0)$.
> - Monotonicity of $D$, $E$: Thank you for suggesting this possible improvement of the paper. We provide **new visualizations** for the curves $D$ and $E$ in practice (see general comment (c) above). For example, as suggested, an example is shown (CMD) where the monotonicity assumption is not fully satisfied since the mean curve of $D(\alpha)/D(0)$ tends to increase slightly for $\alpha\geq 1$ (though there is a high standard deviation). However, in general, there is a clear trend for $D$ and $\bar E$ to **follow the assumptions** and the balancing principle estimate is near to the region where the true balancing value is expected to be. This general trend is also supported by the high performance of the BPDA in our empirical evaluations.
> - Standard deviation: **We disagree** that our method has on average a significantly higher standard deviation than the other methods (IWV and DEV) **for CMD and MMD**. The BPDA has only on 13 tasks (B->D, CS->R, ...) out of total 38 tasks the highest standard deviation for CMD and MMD. From our point of view, this $1/3$ of tasks does not significantly underpin a general trend of a higher standard deviation of the BPDA. But **we agree** that the standard deviation **for DANN** is higher. Please note that the standard deviation for DANN tends to be higher for all methods on all datasets. This behaviour is also evident in the original DomainNet-19 paper [n1] (Table 5; source combine), where DANN has the highest standard deviation of all models.
>
> <sup>[n1] Peng, X., Bai, Q., Xia, X., Huang, Z., Saenko, K., & Wang, B. (2019). Moment matching for multi-source domain adaptation. In Proceedings of the IEEE/CVF International Conference on Computer Vision (pp. 1406-1415).<sup>
> - Computation time: At a first glance, the quadratic number of cross-norm computations seems to consume a lot of time. However, **we disagree** that this is a problem in practice (especially when compared to other model selection methods) due to the following four reasons: (Reason 1) One cross-norm computation needs approximately the same time to compute as one forward pass to train a network. That is, the main time is needed for training the networks. In practice, on the DomainNet dataset, all cross-norm computations together (for 12 models) need 19.2 minutes which is **less than a single network training** for 33 minutes. (Reason 2) One needs more time to compute all the proxy-A distance values (which need to train a classifier for each value) than all the cross-norm computations for one model $f_i$. However, other approaches also need to train additional classifiers. For example, when looking at the original DEV source code (on github), one can observe that even **9 different MLP networks** are trained to estimate the importance weights for a single model. (Reason 3) The activations of higher layers can be cashed and later used for cross-norm computations. This needs **more memory, but also less running time**, as the cross-norm computation reduces to two cashings and one loss computation. (Reason 4) We performed a small real-world comparison regarding computation times. The result is the following: IWV as computed in the original DEV source code (i.e. computing 9 MLPs for the importance weight on the activation space), for 1 model on DomainNet dataset, for 12 $\alpha$ parameters, without repetitions for initializations, sequentially computed on a single core of a single machine (without parallelization), needs approximately 45 hours while the same setup for BPDA needs approximately 59 hours. This is not that much more considering that IWV is the fastest of all compared methods.
> - Remark on stochastic speedup: Indeed, your important remark about stochastic speedup can be **easily realized**. More precisely, the cross-norm values can only be computed for adjacent parameter values $f_i$ and $f_{i+1}$. This leads to a faster method with similar theoretical guarantees (e.g. same optimality of rate). Please see [38] Eq. (11) and Theorem 2 in [38] for details on this linearization. The proof (of Theorem 2 in [38]) can be adapted to the DA setting in the same way as it is done for Theorem 1 in [38] to our Theorem 1. We performed experiments based on adjacent weighting parameters only. Unfortunately, the speedup comes at the cost of worse results for both, theoretical constants in bounds, stronger assumptions, and worse empirical evaluations.
> - Related work: We thank you very much for the **pointers to missing literature** which we will discuss in the related work section (in addition to the Limitations section and other places in the paper where the papers belong to). In particular, we will include the TrCV method whenever methods using density ratio are considered, and, we will present the idea of “circular validation” for both, TrCV and DASVM, in the related work section. In particular, we will mention (in the related work section) that *“...Other important model selection methods are [TrCV and DASVM] which rely on the working hypothesis that the performance of a classifier trained on reversed domains, i.e. source and target switched, gives a good indication for the accuracy of parameter choices….”*
> - Baseline TB: Indeed, we consider the **first case** you describe where **no conflict** is present.
> - Limitations: Please see our **new Limitations section** in the general remark above.
>
> We **thank you again** very much for your constructive comments, help, and the time you invested to review our work! If you are satisfied with our answers and changes, we kindly ask you to take this into account when reconsidering your score.

---

> > ### Comment · Reviewer_DYjR · 2021-08-25
> > **Thanks for the feedback**
> >
> > I would like to thank the authors for the feedback provided, I appreciate it.
> >
> > About the tightness of the bound, I was wrong indeed about D(0), but actually I was more thinking on the importance of $q$ and the fact that it can be large. You answered in your general answer, but I am not sure to understand the remark  "it becomes looser for lower numbers of weights (q is larger)", could you precise this point.
> >
> >
> > About the standard deviation. Actually, I think that the discussion should be related to the datasets considered. For example for Amazon reviews and MMD, the standard deviation  is larger than DEV in more than 50% of the cases. For miniDomainNet with CMD and DEV it is always higher. So, ok the method has a not always a higher standard deviation on average, but it can happen for some tasks/datasets.
> > My question is then: when the method seems less stable, is it an indicator that the assumptions are not completely satisfied?
> >
> > Thanks for your precisions.

---

> > > ### Author Response · Authors · 2021-08-26
> > > **Thank you for your helpful comments**
> > >
> > > We thank you very much for your answer and your helpful comments!
> > >
> > > We are glad that our answers could **clarify all of your concerns except** (a) tightness of our generalization bound w.r.t. the factor $q$, and, (b) higher standard deviation on some tasks. Please find our answers below:
> > >
> > > (a) Tightness of generalization bound: The main message of Thm. 1 is the **optimality of the error rate** which is (to the best of authors’ and reviewers’ knowledge) the **first non-trivial guarantee for a model selection approach under disjoint supports**. The constant of the bound depends on $q$ which can be chosen, by Eq. (14), as the maximum slope between two consecutive weights $\alpha_l$ and $\alpha_{l+1}$, i.e. $q:= \max_l D(\alpha_l)/D(\alpha_{l+1})$. Theoretically (in a pathological example), $D(\alpha_l)$ can be large and $D(\alpha_{l+1})$ can be small for some specific $l$, i.e. $D$ decreases only between two specific $\alpha_l,\alpha_{l+1}$ and is more or less constant for all others. In such examples, the constant of our bound is large (and the bound gets loose) in the same way as the constants in most traditional generalization bounds get large for pathological examples, see e.g. [1,2,3]. Note that this phenomenon disappears for increasing numbers $w\to\infty$ of values $\alpha_1,\ldots,\alpha_w$, since $q\to 1$ in this case. In practice, a geometric series for $\alpha$ leads to good results for the balancing principle. See also [4,5] for a more detailed discussion of the balancing principle behaviour regarding $q$.
> > >
> > > [1] Vapnik, V. (2013). The nature of statistical learning theory.\
> > > [2] Zhang, C., Bengio, S., Hardt, M., Recht, B., & Vinyals, O. (2021). Understanding deep learning (still) requires rethinking generalization. Communications of the ACM, 64(3), 107-115.\
> > > [3] Ben-David, S., Blitzer, J., Crammer, K., Kulesza, A., Pereira, F., & Vaughan, J. W. (2010). A theory of learning from different domains. Machine learning, 79(1), 151-175.\
> > > [4] Pereverzev, S., & Schock, E. (2005). On the adaptive selection of the parameter in regularization of ill-posed problems. SIAM Journal on Numerical Analysis, 43(5), 2060-2076.\
> > > [5] De Vito, E., Pereverzyev, S., & Rosasco, L. (2010). Adaptive kernel methods using the balancing principle. Foundations of Computational Mathematics, 10(4), 455-479.
> > >
> > > (b) Higher standard deviation in some tasks: The **answer to your question is “No”**: A higher standard deviation in some tasks does not indicate violated assumptions. For a counter example consider MMD on Transformed Moons (Table 1 in supplementary and figure at https://anonymous.4open.science/r/bpda-1D3D/figures/monotonicity.jpg). The standard deviation is higher but the assumptions are satisfied.
> > >
> > > From our point of view, a statistically significant **trend of a higher standard deviation is not present**. Even if the standard deviation is higher in some tasks (as you mention), from our point of view this is a rather random phenomenon, not related to the datasets considered. For Amazon Reviews, the mentioned 50% of the cases are mostly cases where our method significantly outperforms DEV (as in most other cases on this dataset). For DomainNet and MMD, the standard deviation is in only 1 (out of 6 cases) the highest. For DomainNet and CMD, yes, the standard deviation is higher, but only **because DEV chooses bad models** (with low variance); DEV is clearly significantly outperformed by our method in all these cases by a large margin.
> > >
> > > We thank you again very much for your constructive comments and scientific discussion! If (some of) your concerns are properly addressed, we kindly ask you to take this into account in your score resetting.

---

> > > > ### Comment · Reviewer_DYjR · 2021-08-27
> > > > **Thanks for the feedback**
> > > >
> > > > Dear authors,
> > > >
> > > > Thanks for your feedback. I appreciated the answers and I agree to increase my score.

---

> > > > > ### Author Response · Authors · 2021-08-27
> > > > > **Thank you for your feedback**
> > > > >
> > > > > Thank you for your time, all the constructive comments and your feedback on our work!

---

### Official Review · Reviewer_VzZx · 2021-07-18

**Rating:** 6
**Confidence:** 3

**Summary:**

The paper proposes the  balancing principle to tackle hyper-parameter selection in unsupervised domain adaptation without requiring target labelled data. By viewing the target error bound as the combination of "  learning errors"  and the distance between source and target domains, the paper designs an algorithm that balances both terms to estimate the regularization parameter.  Generalization guarantee  of the corresponding selected  model is provided. Empirical  evaluations on a simulation dataset and two classical domain adaptation datasets highlight the performances achieved by the proposed method over existing approaches.

**Ethical Concerns:**

The  paper does not raise  any ethical issues.

**Limitations And Societal Impact:**

The paper  adequately metions the broader impact of the tackled unsupervised domain adaptation . Especially the issue related  to the fact existing bias (including societal bias) in source domain data may be lifted to the target domain after adaptation is discussed.

**Main Review:**

- Overall, the paper is well written. In particular, the rationale behind the proposed method is justified. Empirical evaluations support these intuitions and show how they contribute Unsupervised Domain  adaptation (UDA).
- Specifically, the paper tackles the challenging task of model selection in UDA without having access to labelled data from target domain. As such, the addressed problem is of interest for practitioners. To achieve the hyper-parameter  selection in UDA, the paper adapts to DA the balancing principle, a framework used in ill-posed inverse problems.
- The main pursued idea is to estimate the regularization parameter such that the domain-gap distance is balanced with the learning errors occurring in the target error upper-bound. As not all terms in the learning errors are computable, a surrogate is used to attain the balancing. It  consists in computing the cross-errors over $f_i, f_j, j=1,  \cdots, i-1$ on target data and in ensuring that  the cross-errors are lower than a specified bound depending solely on the domain distance. As a model $f_i$ is related to a hyper-parameter $\alpha_i$  (the $\alpha_i$s  are set increasingly), the best hyper-parameter is the maximal of such $\alpha_i$s. Interestingly the procedure is agnostic to the  theoretical target error bound. As a concept  of proof, the paper investigates the use of two existing target error bounds to illustrate the benefit of the approach. Finally the rationale behind the proposed approach is supported by a generalization guarantee.
- The paper  claims that the proposed method, BPDA, may account for label shift (lines  99-102). However  this assertion is not investigated in the experimental evaluations  or supported by a theoretical  analysis. Note that in UDA,  there exists target bound that  handles the label shift (see [a]). Is this kind of bound more suitable to BPDA to deal with generalized shift?
- The considered target error in Equation (4) is  based on the 0-1 loss that offers the nice triangle inequality property (used to establish Theorem 1). However,  in practice when training the pursued model $f$  one rather relies on surrogate function of the 0-1 loss. How this fact impacts the consistency of estimated hyper-parameter? In the same vein does the  margin loss, used in the the target bound (6), satisfy the properties stated lines 66-68?
-  $D(\alpha)$ is assumed  to  be non-increasing. In  practice when training the UDA model in a deep learning setting,  the assumption may not hold true. Is this  detrimental for the proposed UDA model selection?
- The paper should discuss the tightness of the bound (15).
- In the empirical evaluations on the Transformed Moons, it might be interesting to compute and plot $E(\alpha)$, $D(\alpha)$ to  illustrate the ability of  BPDA to retrieve the correct  regime of the  hyper-parameter.
-  Reported results in Table 2  suggest that BPDA is  effective over existing approaches. However, Table 3 related to the more challenging multi-class classification tends  to mitigate that  effectiveness. In  fact,  Table 3 does not show the significance of BPDA as its standard deviations are higher than for the competitors. At best, BPDA is on par with those methods. Can the authors elaborate more  on this? Notice that instead of comparing the methods using the average of the averaged classification errors, the ranking of the methods will be  more significant.
- The performances of BPDA (as for the competitors)  tend to  vary according to the  used domain distance. Is a setting more suitable for BPDA that can be recommended to practitioners?
- Minor comments
* To  ease readability,  I would suggest  to expand earlier in the paper the terms $E(\alpha)$ and DA(\alpha)$ related to UDA.
* Line 127: the upper bound on the sampling error should read $\eta_{t, \mathcal{G}, \delta}$
* Lines 146-147:  it is  not clear why the second term in $\lambda_{\mathcal{H}}$ expression does not depend on $\phi$, the representation mapping
* Equation (7): the representation mapping function $\phi$ is  missing in  the expression of $\hat{d}$.
* To  help the reader, I would propose to specify the terms $A(\alpha)$ and  $S(\alpha)$ (at least in the supplementary material) when presenting the balancing principle for regularized inverse problems.
* Line 210: note that $\phi$ is not a scalar-valued function.

[a] Domain Adaptation with Conditional Distribution Matching and Generalized Label Shift, Remi Tachet des Combes, Han Zhao, Yu-Xiang Wang, Geoffrey J. Gordon. Neurips 2020

After feedback
--------------------
I thank the authors for providing detailed responses to the raised points. I highly appreciate their endeavour to clarify most concerns, especially the thighness of the target bound,  the practical instanciation of the bound or the discussion on the potential limitations of the approach when dealing with label shift. The sole weakness I may point out (ans this was mentioned in the paper)  is that the approach seems not overcome existing methods on more challenging DA datasets. Nevertheless the proposed method is theoretically grounded which is a good point to be mentioned. I increase my score to 6.

**Time Spent Reviewing:**

8

---

> ### Author Response · Authors · 2021-08-09
> **Response to reviewer VzZx**
>
> We **thank you very much** for your time and efforts to review our paper and all the constructive input you provide! We thank you also for acknowledging that our work is **well written**, the **rationale is well explained**, that **the experiments support the intuitions** and show how we **contribute to UDA**. Please find our answers to your concerns below:
> - Covariate shift: **We don’t claim** that the proposed algorithm has no problem with excessive label shifts. We claim that the proposed approach (as given by Definition 2, Lemma 1 and Theorem 1) does not assume covariate shift. All theoretical proofs stated in the supplementary materials allow $l_S\neq l_T$. We thank you for the reference [a] which (e.g. in Section 2) nicely points out how label shift can cause problems. **We agree** that a more explicit discussion of label shift can imporve our work. Therefore, we will discuss label shift in a **new Limitations section** as stated in the general comment above (Risks 1 and 2). Since we will consider label shift very explicitly there, we think its a good way to improve clarity and focus by shortening lines 99-102 as follows: *“In contrast to state-of-the-art methods, the BPDA does not assume a bounded ratio between target and source density (sample selection bias assumption).”*
> - Triangle inequality: Theorem 1 **does not assume** that the source cross-error $\varepsilon_S$ satisfies the triangle inequality. The only requirement (needed in the proofs in supplementary, lines 25 and 39) is that the target cross-error $\varepsilon_T$ satisfies the triangle inequality. You well detected a **sloppy definition in our Notation** paragraph. We will change that (see general comment above). Please note that, by precising the assumption on the target error, no hyperparameter is affected by losses used in the proposed methods. We thank you again for your careful reading.
> - Assumption on $D$: **Yes**, in practice (at least the trend of) $D(\alpha)$ should be non-increasing. Otherwise, the BPDA most likely returns the maximum of all $\alpha_1,\ldots,\alpha_w$. However, this is almost never the case in our empirical evaluations as shown by the tables in the supplementary material. For further empirical evidence regarding the monotonicity of $D$, we provide new visualizations (see (c) general comment above).
> - Tightness of bound: We thank you for your suggestion to discuss the tightness of Theorem 1 in more detail. Please find the planned discussion in the general comment above (point (a) and new section (b), Risk 1 and 2).
> - New Visualizations: Thank you for suggesting this extension of the paper. We provide **new visualizations** as suggested (see (c) in general comment above).
> - Multiclass classification: From our point of view a statistically significant trend of a higher standard deviation of BPDA compared to others **is not present in Table 3**. The average standard deviation of BPDA on Table 3 is always between DEV and IWV. From our point of view this does not significantly underpin a trend of being higher than the others. We **agree** that BPDA is on par with others on DomainNet (which we also mention in the paper) while the BPDA satisfies theoretical guarantees under disjoint supports and outperforms others on Amazon Reviews and Transformed Moons. As you point out, the multiclass problem is a difficult problem setting. This is also underpinned by a higher (compared to Amazon Reviews) standard deviation of all methods.
> - Instantiation of BPDA: Our suggestions for best practice consists of two steps: (a) one should check if the chosen distance tends to be decreasing (as it is the case in the new figure of our general comment); (b) a distance should be chosen which has a comparably low variance. A high variance might aggregate with a high variance of the underlying domain adaptation method which can cause instability issues in practice.
> - Minor comments: We thankfully **correct most of your suggestions**, except: the definition of $\lambda_\mathcal{H}$ which we define as function taking some $\phi$ and returning the error; Eq. (7) which takes $\phi({\bf x})$ and $\phi({\bf x’})$ as inputs (see also the line above) and should, as a distance, work without $\phi$.
>
> We **thank you again** for your constructive comments and the time you invested! If you are satisfied with our answers and changes, we kindly ask you to take this into account in your score decsision.

---

> ### Author Response · Authors · 2021-08-25
> **2nd response to reviewer VzZx (after review update)**
>
> Thank you for updating your review with a response to our rebuttal! We are glad about our consensus in the following points:
> * Our approach **sheds new light** on the estimation methodology to deal with "inaccessible" in the common domain adaptation bounds---the joint error minimizer.
> * Our approach is (to the best of the authors' and the reviewers' knowledge) the **first parameter choice method** with non-trivial (optimal error rate) guarantees **under the common practical setting of disjoint supports**.
> * Our approach **significantly outperforms others** on Transformed Moons and Amazon Reviews for several domain adaptation methods (CMD, DANN, MMD).
> * Our approach **outperforms others in average accuracy** on DomainNet. In particular, our approach has a significantly higher performance than the state-of-the-art approach DEV [1] for CMD.
>
> Regarding your concern: *“The sole weakness I may point out (and this was mentioned in the paper) is that the approach seems not to overcome existing methods on more challenging DA datasets.”*, please find our answer as follows:
>
> **We disagree** that our method does not outperform others on challenging tasks. Indeed, we consider the setup of Amazon Reviews very challenging for model selection (state-of-the-art model selection methods, three different domain adaptation methods, 12 different parameter choices and 12 different domain adaptation tasks) and, here, **the outperformance is significant**. On DomainNet, our approach has a **significantly higher performance** than the state-of-the-art approach DEV [1] for CMD and **outperforms all approaches in average accuracy** for the rest of the methods.
>
> **Thank you again** for your time, feedback and comments! Since we clarified all of your concerns (except one) we kindly ask you to reflect this in a score update.
>
>
> __
>
> [1] You, K., Wang, X., Long, M., & Jordan, M. (2019, May). Towards accurate model selection in deep unsupervised domain adaptation. In International Conference on Machine Learning (pp. 7124-7133). PMLR.

---

### Official Review · Reviewer_9iYu · 2021-07-20

**Rating:** 7
**Confidence:** 4

**Summary:**

This paper introduces a new error bound for the target domain motivated from Tikhonov-regularized inverse problem. This adoption gives an optimal selection scheme for the error bound of target domain based on distance-regularized domain adaptation, which is a theoretical principle for most domain adaptation methods. The proposed method is evaluated on the three benchmarks on domain adaptation.

**Limitations And Societal Impact:**

### related works

Could discuss the your method with the work [R1]? In line 99~102, it seems that covariate shift problem assumption is adopted the other papers. case. Could you refer to papers that take this as an assumption for the method? As I know it is not the case for most of the works. The work [R1] indicates covariate shift is the main problem of domain adaptation solver that relies on minimizing the distance between source and target domains.

### DANN and regularization scheduling

It is common to use domain adaptation parameter $\lambda$ is initialized at $0$ and gradually increased to 1 using the scheduling policy.
$\lambda_p = \frac{2}{1 + exp ( -\gamma \cdot p )} -1$

[R1] On Learning Invariant Representation for Domain Adaptation, ICML 19

**Main Review:**

My concerns are solved by the author. I'm keeping my rating 7 and increasing my confidence from 3 to 4.

==========================================================================================

### Originality

The introduced formulation gives an alternative optimal target error bound for unsupervised domain adaptation. Inspired by the Tikhonov-regularized inverse problem, the authors explore similarities between two problems and leads to solutions for the domain adaptation.

### Quality

I'm not familiar with the Tikhonov-regularized inverse problem, the other parts of the paper seem to be good and written in a good manner. But, work on the limitation of the domain adaptation is missing. I will discuss it in the limitation section below.

### Clarity

The paper is well-structured and easy to follow. Section 2 to 4 gives clever structure to understand the statement.

### Significance

The work introduces a novel scheme to select the optimal regularization parameter for target-error bound for domain adaptation, while there exist many works on domain adaptation giving good practices. The experiment is relatively weak, but it is of acceptable quality. My concern as a practitioner for domain adaptation is this work does not talk about the regularization weight of domain adversarial network (DANN), which is epoch-based scheduling value not a fixed value. I will discuss in the limitation section.

**Time Spent Reviewing:**

8

---

> ### Author Response · Authors · 2021-08-09
> **Repsonse to reviewer 9iYu**
>
> We **thank you very much** for your efforts and time to review our paper! We further thank you for acknolwedging that the **error bound is new** and for **acceptance of our empirical evaluations**. Our rebuttal is as follows:
> - Limitations (covariate shift and related works): Thank you for your constructive suggestion! We provide a **new Limitations section** (see general comment above). Specifically, in point (b) (Risk 1), we discuss that our method does not assume covariate shift and how excessive label shift (violation of covariate shift) can affect our method. In Risk 3, we discuss how label shift can affect domain adaptation methods and discuss related works which can be used to overcome this problem.
> - DANN schedule: We **are using the schedule you are pointing to**. Please see the source code of dn_bp.py at line 16. As you point out, this schedule is proposed for the DANN algorithm in Section 5.2.2 of [15] which is concerned with image classification on deep learning. However, in the same paper [15], the Amazon Reviews dataset is considered in Section 5.1.3 and there, the authors select the weighting parameter out of different values by mentioning in [15, Section 5.1.3]: *“...the adaptation parameter $\lambda$ is chosen among $9$ values...”*. For image classification tasks, the polynomial schedule is multiplied by the weighting parameter $\lambda$ which is fixed to $1$ in [15]. This leads to a gradual change from $0$ to $\lambda$ during training ($\alpha$ in our work). Therefore, there is **no contradiction** to the original DANN paper [15]. For clarification, we **will add** the explanation above to Section 5.2 in the supplementary material.
> - Missing discussion of DANN schedule: In the final paper, **we will address your concern** on the significance by addressing the learning schedule of DANN also **in the Introduction**. More precisely, we will mention that a common approach is to “...multiply a fixed weighting parameter, e.g. $1$ in [15], by a heuristically chosen schedule for changing this parameter during training...”
>
> We **thank you again** for your constructive comments, your efforts and the time you invested! If you are satisfied with our answers and changes, we kindly ask you to take this into account in your final decision.

---

### Author Response · Authors · 2021-08-09
**General response to all reviewers**

We **thank all reviewers** very much for all the invested time, efforts and constructive comments which help to improve our work! We are especially happy about the following.
* There is a clear consensus between the authors and reviewers that the work is **original, clearly presented, and relevant** to the community.
* Our approach is (to the best of the authors' and all the reviewers' knowledge) the **first parameter choice method** with non-trivial theoretical guarantees (optimal error rate) under the **common setting of disjoint supports**.
* We especially appreciate all positive comments on the mathematical foundations, our consistency, and, reviewers' acceptance of our empirical evaluations.

The reviewers suggest enhancements mainly regarding (a) discussion of the tightness of the error bound of Thm. 1 (reviewers DYjR, VzZx), (b) explicit discussion of limitations such as expected under label shift (9iYu, DYjR, VzZx), (c) visualization of $D$ and $E$ in practice (VzZx, DYjR), and, (d) technical issues which can be solved by smaller changes of text (mWtP, 9iYu, DYjR, VzZx). We plan to address the suggestions (a--d) by changes in the paper as follows.

##### **(a) Tightness of bound**
Theorem 1 provides the **optimality of the rate** of the error $\varepsilon_T(g_{\alpha_\mathrm{BP}}\circ\phi_{\alpha_\mathrm{BP}})$ if the minimum $\inf_{\alpha\in[0,\infty)} D(\alpha)+\overline{E}(\alpha)$ is achieved, i.e. the error is only a constant factor $C_{D(0),B,q}$ away from the minimum $\inf_{\alpha\in[0,\infty)} D(\alpha)+\overline{E}(\alpha)$. If $E$ is non-decreasing (which can be assumed since it contains the source error inversely penalized by $\alpha$) then $\overline E =E$ and the BPDA minimizes the instantiation bound at an optimal error rate. We will add the following discussion after the theorem: *“The constant factor is given by $C_{D(0),B,q}:=3 q \max \left[\frac{D(0)}{B},1\right]+\frac{3 B}{D(0)} q \max \left[\frac{D(0)}{B},1\right]$. This constant and the bound in Thm. 1 is larger for steeper D between two consecutive weights and it becomes looser for lower numbers of weights ($q$ is larger).“*

##### **(b) Limitations**
We plan a new Section 6 titled **Discussion of limitations and risks** with the following text.
*“A constructive discussion with anonymous reviewers resulted in the following list of risks and limitations of the proposed approach:*
- *Risk 1 (Label shift): In this work, we do not assume a unique labeling function $l_S=l_T$ for source and target domain (covariate shift assumption), but $l_S$ and $l_T$ should be similar. Even if the labeling functions are different $l_S\neq l_T$, the balancing value $\alpha^\ast$ is well estimable by the balancing principle estimate $\alpha_\mathrm{BP}$ (Definition 2) as proven by Lemma 1. Nevertheless, the quality of the balancing value $\alpha^\ast$ itself can (theoretically) be negatively affected by label shift. For example, the function $E$ can first increase and then decrease (caused by label shift) which implies an increasing difference between $\bar E(\alpha)$ and $E(\alpha)$ for increasing $\alpha$. In such situations, the target error of the model $g_{\alpha^\ast}\circ\phi_{\alpha^\ast}$ could be high and consequently also the one of the balancing principle model $g_{\alpha_{\mathrm{BP}}}\circ\phi_{\alpha_{\mathrm{BP}}}$. However, this is a rather pathological case as $E$ contains the source error which typically increases for increasing $\alpha$ (similarly as $D$ is assumed to decrease for increasing $\alpha$).*
- *Risk 2 (Loose target bound for instantiation): Our model is agnostic w.r.t. the property that it can be instantiated with different target error bounds. However, it is known in inverse problem literature [37,38] that a loose bound can lead to a low performance of the balancing principle. This also holds for the BPDA. Risk 2 can be approached by choosing target error bounds that take into account the specific domain shift situation, e.g. the bound [a] is suitable for general domain shift scenarios (see Risk 1).*
- *Risk 3 (Low performance of all models): Situations exist which hinder distance-regularized domain adaptation methods to perform well. One such situation is label shift, see Risk 1, [R1] and references therein. Such scenarios can cause all models $f_1,\ldots,f_w$ to be inaccurate or unstable. The BPDA will select a model $f_i\in \{f_1,\ldots,f_n\}$ with low target error $\varepsilon_T(f_i)$ compared to other models from $\{f_1,\ldots,f_n\}$. Nevertheless, in such situations, $f_i$ might have a high target error $\varepsilon_T(f_i)$ or it might be unstable. As a solution, distance-regularized domain adaptation methods can be applied with modifications, see e.g. [a,R2,R3,R4].*
- *Limitation 1 (Focus on weight parameter): Our theoretical guarantees and the high empirical performance come at the cost of focusing the selection process on the distance-weighting parameter $\alpha$. This is in contrast to other model selection methods, such as [3,23,22,m3,m4], which can select different types of hyperparameters. This limitation can be approached by employing ideas from multipenalty regularization of inverse problems to combine the BPDA with related approaches [m1]. One approach is to explore a grid of values of several hyperparameters by applying the balancing principle w.r.t. one of them and allowing others to take all corresponding grid values. As the result of such an application, one obtains a much reduced set of hyperparameter combinations of interest and one can apply the balancing principle or related approaches w.r.t. the other parameters, see e.g. [m2] for a recent application of this approach.*

[a] *Domain Adaptation with Conditional Distribution Matching and Generalized Label Shift, R Tachet des Combes, H Zhao, Y-X Wang, G J Gordon. Neurips 2020*
[R1] *On Learning Invariant Representation for Domain Adaptation, ICML 19.*
[R2] *Azizzadenesheli, K., Liu, A., Yang, F., & Anandkumar, A. Regularized Learning for Domain Adaptation under Label Shifts, ICLR 2019.*
[R3] *Le, T., Nguyen, T., Ho, N., Bui, H., & Phung, D. (2021). Lamda: Label matching deep domain adaptation. In International Conference on Machine Learning (pp. 6043-6054).*
[R4] *Wu, Y., Winston, E., Kaushik, D., & Lipton, Z. (2019). Domain adaptation with asymmetrically-relaxed distribution alignment. In International Conference on Machine Learning.*
[m1] *Fornasier, M., Naumova, V., & Pereverzyev, S. V. (2014). Parameter choice strategies for multipenalty regularization. SIAM Journal on Numerical Analysis, 52(4), 1770-1794.*
[m2] *Krasnoschok, M., Pereverzyev, S., Siryk, S. V., & Vasylyeva, N. (2020). Determination of the fractional order in semilinear subdiffusion equations. Fractional Calculus and Applied Analysis, 23(3).*
[m3] *Erheng Zhong, Wei Fan, Qiang Yang, Olivier Verscheure, Jiangtao Ren: Cross Validation Framework to Choose amongst Models and Datasets for Transfer Learning. ECML 2010*
[m4] *Lorenzo Bruzzone, Mattia Marconcini: Domain Adaptation Problems: A DASVM Classification Technique and a Circular Validation Strategy. TPAMI 32(5): 2010."*

##### **(c) Visualizations of curves in practice**
We performed new visualizations for $D$, $E$ and $\bar E$ on the Transformed Moons dataset. We estimated $\lambda_{\mathcal{G}}(\phi)$ by training a classifier similar to the proxy-A distance. The new figure below compares the results with the theoretical assumptions for the methods CMD and MMD.

[[Anonymous and save link to figure]](https://anonymous.4open.science/r/bpda-1D3D/figures/monotonicity.jpg)

The following text and the figure will be added to the paper in Section 5.2: *“Figure 3 shows the behaviour of the curves $D(\alpha)/D(0)$, $E(\alpha)/B$ and $\bar E(\alpha)/B$ for the two methods CMD and MMD and the Transformed Moons dataset. The following observations can be made.*
- *$E$ is bounded, the mean curves of $D(\alpha)$ tend to be non-increasing and the technical assumption $\frac{D(\alpha)}{D(0)}\leq \sup_{\alpha\in[0,\infty)}\frac{E(\alpha)}{B}$ is satisfied for $\alpha\geq 10^{3}$. That is, the mean curves tend to follow the assumptions listed in Section 4.1.*
- *The mean curves $\bar E$ and $E$ tend to be similar. That is, Risk 1 does not apply.*
- *The average parameters chosen by the balancing principle (see $\alpha_\mathrm{BP}$ in Figure) are the maximum values for which the mean curves of $E(\alpha)/B$ are smaller than the mean curves of $D(\alpha)/D(0)$. That is, the BPDA described in Algorithm 1 tends to perform as expected.*
- *There is a small trend towards a violation of the monotonicity assumption for $D$ in the right subfigure for CMD. However, the BPDA is (on average) robust w.r.t. this violation as it picks nearly the optimal value, see Table 2 and the supplementary material.”*

##### **(d) Technical issues requiring only smaller changes**
We further acknowledge the support of the reviewers to identify issues which require minor text changes:
- The polynomial schedule for the weighting parameter in [15] will be mentioned in the third paragraph of the introduction by “...multiply a fixed weighting parameter, e.g. $1$ in [15], by a heuristically chosen schedule for changing this parameter during training...”
- Only the target errors need to satisfy the triangle inequality (paragraph Notation). The mentioned general focus on errors satisfying the triangle inequality is wrong. Source errors are allowed to violate the triangle inequality.
- We will define $D(\alpha)$ and $E(\alpha)$ more concretely earlier in the text.
- Line 127: $t$ instead of $z$.
- We will give examples of the sampling error $S$ and the approximation error $A$ of some inverse problem in the supplementary material for completeness.
- $\mathcal{G}$-divergence will be replaced by $d_{\mathcal{G}\Delta\mathcal{G}}$-divergence.
- The numbers of the Lemmas in the supplementary material will be aligned with the main text.

---

### Decision · Program_Chairs · 2021-09-27

**Decision:**

Accept (Poster)

**Comment:**

The submission proposes the balancing principle to tackle hyper-parameter selection in unsupervised domain adaptation, with an error bound for the target domain.  The reviewers were unanimous that the paper was above the acceptance threshold for NeurIPS.  In the post rebuttal discussion, it was pointed out that the method is supported by theoretical analysis, and although it is not likely that this current paper will be the final word on model selection for domain adaptation, that it provides a concrete step forward in a topic that has a lack of theoretical analysis.  As such, it is of interest to the NeurIPS community and may foster additional research and analysis on the topic.